# ReMoE: Boosting Expert Reuse through Router Fine-Tuning in Memory-Constrained MoE LLM Inference

**Xiongwei Zhu** [1]  **Xiaojian Liao** [1]  **Tianyang Jiang** [2]  **Yusen Zhang** [1]  **Liang Wang** [1]  **Limin Xiao** [1]

## Abstract

Fine-grained Mixture-of-Experts (MoE) models sparsely activate only a subset of experts per token, reducing activated computation while maintaining high model capacity. However, in memory-constrained inference scenarios, only a small set of experts can be cached. Experts not in the cache must be fetched from slow external storage (e.g., UFS), leading to frequent evictions and substantial I/O overhead. We propose ReMoE, a router fine-tuning framework designed to boost token-wise expert reuse. ReMoE biases the router toward recently selected experts, producing temporally stable routing that better matches cache locality constraints. By increasing short-horizon expert reuse, ReMoE reduces expert fetches from storage without adding inference-time computation. Experiments on DeepSeek and Qwen models show that ReMoE improves expert reuse by 26% while maintaining downstream task performance. Real-system evaluations further confirm these benefits, improving output throughput by 8.4% under vLLM GPU–CPU expert offloading and reducing TPOT by 43.6–49.8% under llama.cpp on Jetson Orin NX, corresponding to a 1.77–1.99× decode speedup across diverse workloads. Checkpoints and usage instructions are available at https://github.com/BUAA-OSCAR/ReMoE.

## 1. Introduction

Edge-side deployment of Large Language Models (LLMs) is an emerging trend (Xu et al., 2024; Zheng et al., 2025; Wang et al., 2025), with on-device AI applications such as real-time translation and advanced photo editing already

---

[1]School of Computer Science and Engineering, Beihang University, Beijing 100191, China [2]Huawei Technologies Ltd. Correspondence to: Xiaojian Liao <liaoxj@buaa.edu.cn>.

*Proceedings of the 43$^{rd}$ International Conference on Machine Learning*, Seoul, South Korea. PMLR 306, 2026. Copyright 2026 by the author(s).

demonstrating its practical value (Xue et al., 2024). While Mixture-of-Experts (MoE) architectures are central to scaling LLM capacity, their deployment has so far remained predominantly cloud-based, with limited use on edge devices. Recent efforts on on-device MoE models, edge-oriented inference runtimes, and mobile MoE applications suggest that this situation is beginning to change, making MoE increasingly relevant to edge and memory-constrained deployments (OPPO, 2024; MediaTek, 2024; Nvidia, 2026; Liquid AI, 2025; Ai2, 2025; Google DeepMind, 2026).

This trend is supported by two factors. First, MoE's sparse activation keeps only a subset of experts active during inference, preserving large model capacity while limiting the activated parameter footprint. Second, advances in mobile storage make local weight storage increasingly feasible. For example, Samsung's UFS 4.0 offers read speeds up to 4 GB/s and capacities up to 1 TB, making local storage of large model weights increasingly feasible (Samsung Semiconductor). However, deploying MoE LLMs on edge devices introduces the challenge of frequent expert switching. During the decoding phase, where each token may activate a different set of experts, this results in frequent, irregular I/O requests to load the required expert weights from storage, prolonging the inference latency (Qu et al., 2025).

Current solutions typically attempt to hide this latency through system-level techniques such as prefetching or caching algorithms. A key upstream factor is the routing trace produced by the MoE router, i.e., the sequence of selected expert sets $\{E_t\}_{t=1}^{T}$ across decoding steps. Many standard MoE training recipes include load-balancing objectives that spread tokens across experts to improve training-time utilization under expert parallelism. While useful for large-scale training, such dispersion can be misaligned with memory-constrained single-request decoding, where inference benefits when adjacent tokens reuse part of the same expert working set. ReMoE addresses this training–deployment mismatch by shaping the routing trace itself, complementing runtime caching and prefetching.

To bridge this gap, we propose ReMoE, a lightweight router fine-tuning framework that aligns routing behavior with memory capacity constraints on edge devices. ReMoE adapts the router using two complementary objectives: (i) a

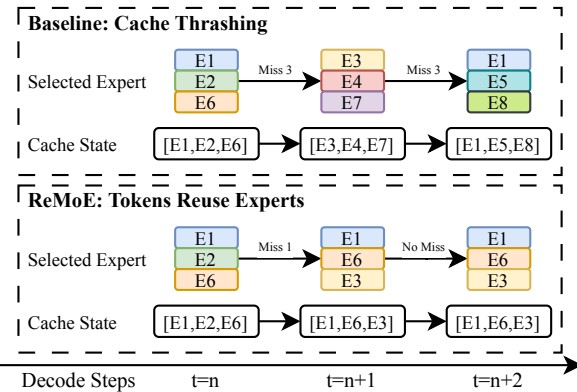

*Figure 1.* **Comparison of expert I/O dynamics. Baseline:** Standard routers select disjoint experts across steps, causing frequent cache replacements and I/O thrashing. **ReMoE:** Our method encourages temporal locality by increasing expert reuse across adjacent steps, thereby reducing cache turnover and I/O overhead.

temporal locality loss that encourages expert reuse across adjacent tokens, and (ii) a Trust-KL loss that softly anchors the updated routing distribution to the pretrained router. This biases the router toward temporally stable reuse while constraining distributional drift, transforming fragmented routing traces into more cache-friendly sequences. Crucially, ReMoE does not modify the model architecture, hardware, or inference kernels, and adds no runtime policy beyond using the fine-tuned router weights.

We evaluate ReMoE across fine-grained MoE models and heterogeneous serving platforms. ReMoE increases expert overlap by 26.4% on DeepSeek-V2-Lite and by 27.2% on Qwen1.5-MoE-A2.7B. We further perform trace-driven offline cache analysis using unique hit rate (uHR), the cache-hit fraction over distinct expert requests, and total unique misses (#uMiss), the number of distinct expert loads. ReMoE improves uHR and reduces #uMiss across cache capacities and replacement policies. In real-system evaluations, ReMoE improves output throughput by 8.4% and reduces TPOT by 4.5% under vLLM GPU–CPU expert offloading. On Jetson Orin NX, where expert misses are more expensive under SSD-backed edge inference, ReMoE reduces TPOT by 43.6–49.8% across ShareGPT, GSM8K, and HumanEval, corresponding to a 1.77–1.99× decode speedup.

## 2. Related Work

**MoE routing, load balancing, and locality.** Mixture-of-Experts (MoE) scales model capacity by activating only a few experts per token (Shazeer et al., 2017; Lepikhin et al., 2021; Fedus et al., 2022; Du et al., 2022). Most MoE models use Top-$K$ token-choice routing with auxiliary balancing losses to prevent collapse and improve utilization (Lepikhin et al., 2021; Fedus et al., 2022). Recent work explores alternative routing objectives and mechanisms, such as router

z-loss for stability (Zoph et al., 2022), expert-choice routing for better balance and efficiency (Zhou et al., 2022), and auxiliary-loss-free balancing strategies (Wang et al., 2024). Prior analyses study when learning to route helps and how routing variants affect quality (Dikkala et al., 2023), with surveys summarizing modern MoE routing and training practices (Cai et al., 2025). Fine-grained MoEs, including DeepSeek-V2/V3 and Qwen MoE, increase specialization but also amplify token-wise expert switching (DeepSeek-AI et al., 2024; Qwen Team, 2024; DeepSeek-AI et al., 2025; Yang et al., 2025). Oracle-MoE addresses the resulting locality problem by redesigning the routing architecture and training from scratch to preserve expert activation consistency (Zhou et al., 2025). ReMoE targets the same locality bottleneck from a different angle: rather than modifying the architecture or requiring pretraining, it fine-tunes only the existing gate parameters of an already-trained MoE checkpoint, making it a lightweight post-training adaptation that leaves the model architecture and expert weights unchanged.

**System support for memory-constrained MoE inference.** System efficiency for MoE inference depends on dispatch, communication, expert parallelism, and expert-weight movement; frameworks such as DeepSpeed-MoE and Tutel/MegaBlocks optimize dispatch and expert-parallel execution costs (Rajbhandari et al., 2022; Hwang et al., 2023; Gale et al., 2023; Lin et al., 2025). Under memory constraints, MoE offloading systems reduce expert-transfer overhead through caching, offloading, and CPU–GPU orchestration, as shown by MoE-Infinity, HOBBIT, FineMoE, KTransformers, MoE-Lightning, and Fiddler (Xue et al., 2025; Tang et al., 2024; Yu et al., 2025; Liang et al., 2025; Chen et al., 2025a; Cao et al., 2025; Kamahori et al., 2025). CoServe shows that expert-based collaborative inference also suffers from memory-tier switching overhead, motivating dependency-aware scheduling and expert management (Suo et al., 2025). Some cache-aware routing methods, such as Mixture of Cache-Conditional Experts, bias expert selection using cache residency at inference time, directly trading off cache hits and routing choices during decoding (Skliar et al., 2025). ReMoE is complementary to these runtime methods: it reshapes the routing trace offline through router fine-tuning, so the deployed model can use the same inference graph and standard cache policies. Although related memory-constrained serving systems have also been studied for dense LLMs (Sheng et al., 2023; Alizadeh et al., 2024; Xue et al., 2024; Jiang et al., 2024; 2025; Du et al., 2025; Bian et al., 2025), ReMoE focuses on the MoE-specific problem of token-wise expert switching and cache locality. At the storage layer, prior systems improve I/O efficiency through write-dependency disentanglement, multicore flash-file-system scalability, and crash-consistent NVMe support (Liao et al., 2020; 2021a;b); these optimizations are complementary to ReMoE, which reduces the up-

stream expert-access demand generated by the router.

**Deployment-aware training and compression.** A common principle is to incorporate deployment constraints during training so inference remains simple, such as latency-/hardware-aware pruning and architecture optimization (Shen et al., 2022; Kurtic et al., 2023). For model compression, quantization reduces memory footprint and bandwidth demand through post-training quantization and low-bit inference methods such as SmoothQuant, GPTQ, AWQ, and ZeroQuant, as well as low-bit fine-tuning methods such as QLoRA (Xiao et al., 2024; Frantar et al., 2023; Lin et al., 2024; Yao et al., 2022; Dettmers et al., 2023). Recent fine-grained quantization and algorithm–accelerator co-design methods further mitigate outliers and salient weights to improve low-bit LLM inference efficiency (Xie et al., 2025; Xie et al., 2026). Quantization-aware training has also become practical for improving low-bit quality under deployment constraints (Liu et al., 2024; Chen et al., 2025b; Esser et al., 2025). ReMoE follows the deployment-aware principle at the router level: it freezes all non-router parameters and fine-tunes only the gate to encourage short-horizon expert reuse, aligning routing with cache locality without added inference-time computation. Because expert weights remain frozen, ReMoE is orthogonal to parameter-space optimizations: improved reuse reduces how often experts are fetched, while low-bit experts reduce the cost of each fetch.

## 3. The Locality Gap in MoE Offloading

We study fine-grained MoE decoding under memory-constrained, single-request inference on edge devices, where fast memory is limited. Since modern MoE LLMs can require tens of GB for weights alone, while edge DRAM must also accommodate runtime buffers and KV cache, only a small subset of experts can remain resident in fast memory. The remaining experts must be fetched from slower storage (e.g., UFS) on demand, making expert-weight movement a first-order bottleneck. In contrast to datacenter serving, this regime typically operates with $B=1$ (interactive usage), leaving little opportunity to amortize I/O latency across batches. For clarity, our cache analysis adopts a request-isolated setting where each prompt starts from a cold expert cache. As a result, the step-to-step stability of routed experts becomes a primary determinant of end-to-end latency.

### 3.1. Preliminaries and Notation

**MoE routing formulation.** An MoE layer contains $N_r$ routed experts $\{e_i\}_{i=1}^{N_r}$ and a router with parameters $\theta_{\text{gate}}$. Given hidden state $h_t$, the router computes $P_t = \text{Softmax}(h_t^\top \theta_{\text{gate}})$ and selects $E_t = \text{Top-K}(P_t)$ with $|E_t| = K$. Shared experts (if present) are always activated and are excluded from $P_t$.

*Table 1.* **Key notation.** Shared experts are excluded from $P_t$.

| Symbol | Meaning |
|---|---|
| **Baseline MoE / Inference & Cache** (Sec. 3) | |
| $t, T$ | decoding step; total steps |
| $h_t$ | hidden state at step $t$ |
| $N_r$ | routed experts per MoE layer |
| $P_t \in \mathbb{R}^{N_r}$ | routing distribution over routed experts |
| $E_t$ | selected expert index set |
| $C$ | per-layer expert cache capacity |
| $\text{IR}_t$ | Instantaneous Reuse |
| $\text{EOR}$ | Expert Overlap Ratio |
| **ReMoE / Router Fine-Tuning** (Sec. 4) | |
| $\theta_{\text{gate}}$ | trainable gate parameters |
| $\theta_{\text{gate}}^0$ | frozen snapshot of pretrained gate |
| $P_t^{\text{ref}}$ | reference distribution from $\theta_{\text{gate}}^0$ |
| $L_{\text{Trust}}$ | KL anchor between $P_t$ and $P_t^{\text{ref}}$ |
| $L_{\text{Loc}}$ | temporal locality regularization |
| $\lambda_{\text{KL}}, \{\lambda.\}$ | weights for trust/locality terms |

**Inference setting.** We consider autoregressive decoding for a sequence of length $T$ under memory-constrained, single-request inference ($B=1$). We focus on expert-granularity weight movement and model a per-layer expert cache of capacity $C$ (experts/layer) in fast memory. At step $t$, the requested working set is $E_t$; cache hits avoid weight loads, while misses trigger expert fetches from storage.

### 3.2. Metrics for Offloading Efficiency

We quantify how cache-friendly a routing trace is $\{E_t\}_{t=1}^{T}$, without assuming a particular caching or prefetching policy. Since decoding is sequential and cache state evolves over time, we evaluate reuse at the level of adjacent steps.

**Instantaneous reuse and Expert Overlap Ratio (EOR).** We quantify short-horizon routing locality by the step-to-step overlap

$$\text{IR}_t = \frac{|E_t \cap E_{t-1}|}{K}, \qquad \text{EOR} = \frac{1}{T-1}\sum_{t=2}^{T} \text{IR}_t, \quad (1)$$

where larger values indicate stronger expert reuse across adjacent decoding steps, implying fewer on-demand expert fetches under a cache.

**Proposition 3.1.** *Consider a per-layer expert cache of capacity $C \geq K$ with a recency-based replacement policy (e.g., LRU) and request-isolated decoding. Let $E_t$ be the Top-$K$ routed expert set at step $t$. Then the number of expert fetches at step $t$ satisfies $N_{\text{fetch}}(t) \leq K - |E_t \cap E_{t-1}| = K(1 - \text{IR}_t)$, and thus the average fetches satisfy $\bar{N}_{\text{fetch}} \leq K(1 - \text{EOR})$.*

A proof is provided in Appendix A, which also lists concrete failure modes when the assumptions break. We treat

this result as motivating analysis: it shows EOR is a meaningful proxy for I/O efficiency under standard caching semantics, but does not directly constrain ReMoE's optimization—which operates on differentiable distribution-level surrogates (Sec. 4)—and the primary validation comes from experiments (Sec. 5).

### 3.3. The Training–Inference Mismatch

Standard MoE training often uses an auxiliary load-balancing loss $L_{\text{aux}}$ to spread tokens across experts for expert-parallel training. This objective can conflict with memory-constrained single-request decoding, where reusing a compact expert working set reduces expert fetches. Figure 2 shows that the baseline router has short reuse streaks but frequent step-to-step switching, indicating exploitable natural locality. ReMoE targets this mismatch by increasing short-horizon overlap while anchoring the router to the pretrained distribution. A moderate increase in inference-time routing imbalance is acceptable in our $B{=}1$ setting because there is no expert parallelism to protect, and local concentration directly reduces distinct expert loads.

## 4. ReMoE: Internalizing Expert Cache Locality via Router Fine-Tuning

ReMoE reshapes MoE routing to be more cache-friendly without modifying expert parameters or the inference runtime. We freeze all non-router weights—including embeddings, attention blocks, and expert FFNs—and fine-tune only the gate parameters $\theta_{\text{gate}}$. As a result, ReMoE is lightweight to train and preserves the baseline inference graph, incurring zero runtime overhead at deployment.

**Scope and indexing.** We apply the same objective to every MoE gate in the model and average the losses across MoE layers and token positions. We use teacher forcing during fine-tuning, so the input token sequence and the time index $t$ are fixed; however, the hidden states $h_t$ are still produced by the current model (and can change as routing changes). Our regularizers operate directly on router outputs $\{P_t\}$, encouraging temporally local routing while keeping the router semantically anchored to the pretrained behavior.

### 4.1. Overview

Figure 3 illustrates ReMoE within a single MoE layer. Given the token hidden state $h_t$, we run a frozen reference router and a trainable router in parallel to obtain $P_t^{\text{ref}}$ and $P_t$ (Step **1**). Concretely, we store a frozen FP32 snapshot of the pretrained gate weights $\theta_{\text{gate}}^0$ and compute $P_t^{\text{ref}} = \text{Softmax}(h_t^\top \theta_{\text{gate}}^0)$, while updating only $\theta_{\text{gate}}$. We then optimize the trainable router with two signals (Step **2**): (i) a semantic anchor that keeps $P_t$ close to $P_t^{\text{ref}}$, and (ii) a temporal locality signal that relates $P_t$ to a short routing his-

tory. Only the trainable gate receives gradients (Step **3**). We maintain a small history buffer of recent routing outputs (or the corresponding Top-$K$ sets) to construct locality targets for subsequent steps (Step **4**). Finally, the selected Top-$K$ experts execute exactly as in the baseline (Step **5**).

**From cache locality to a router objective.** Our goal is to reduce expert offloading under a small per-layer cache. As shown in Sec. 3.2, step-to-step overlap (EOR/IR) provides an upper bound on fetches under recency-based caching, motivating higher adjacent-step reuse. ReMoE reshapes the routing trace and is thus complementary to cache replacement and prefetching, which can handle residual long-tail misses beyond capacity $C$.

However, the discrete Top-$K$ operator $E_t = \text{Top-K}(P_t)$ is non-differentiable, so we optimize a smooth surrogate based on the reuse mass that $P_t$ assigns to previously selected experts.

Let $\tilde{E}_{t-1} = \texttt{stop\_gradient}(E_{t-1})$ denote the previous-step routed set treated as a constant. The $\texttt{stop\_gradient}$ operator treats the previous Top-$K$ set as fixed, so gradients flow only through the current routing distribution $P_t$. This provides a one-way reuse signal: the current step is encouraged to reuse the previously realized expert set.

We then define the reuse mass as

$$ m_t = \frac{1}{K} \sum_{k \in \tilde{E}_{t-1}} P_t^{(k)}. \tag{2} $$

A larger $m_t$ means $P_t$ assigns higher probability to experts that were activated at step $t{-}1$, which increases the likelihood that the next routed set $E_t$ overlaps with $E_{t-1}$. ReMoE further combines this surrogate with smoothness/inertia/working-set terms (Sec. 4.4) to suppress both high-frequency jitter and slow drift in routing trajectories. Further justification for optimizing reuse mass as a differentiable surrogate for step-to-step Top-$K$ overlap is provided in Appendix B.

### 4.2. Training Objective

**Balancing locality and semantic drift.** Our goal is to improve cache locality while preserving the routing semantics learned during pretraining. We therefore optimize a single weighted objective: temporal-locality regularizers encourage reuse, and a KL penalty to a frozen snapshot router acts as a soft trust-region that limits semantic drift. This design needs neither a separate teacher model nor any inference-time modifications.

We keep the base causal language modeling signal and add router-specific regularization. Let $L_{\text{CE}}$ be the standard next-token cross entropy loss. During router fine-tuning, we disable the standard MoE load-balancing loss ($L_{\text{aux}}{=}0$) be-

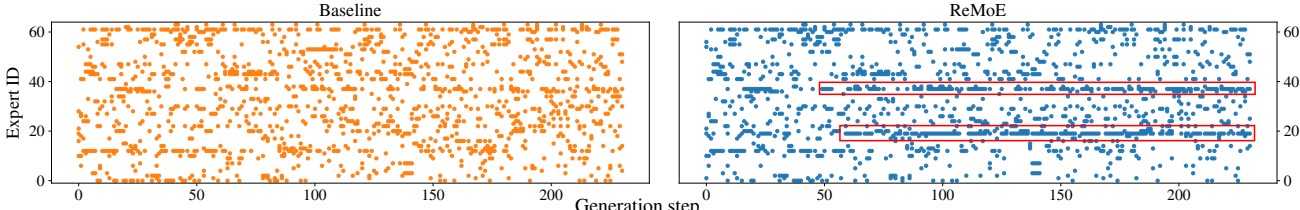

*Figure 2.* **Routing trajectories under teacher forcing (21st MoE layer of DeepSeek-V2-Lite).** We trace Top-$K$ expert indices over decoding steps for a fixed token sequence. The baseline already shows short stretches of potential reuse but exhibits frequent switching, while ReMoE—via gate-only fine-tuning—extends these streaks and stabilizes the working set, increasing short-horizon overlap. With inputs fixed, the difference reflects a change in routing policy rather than generation divergence.

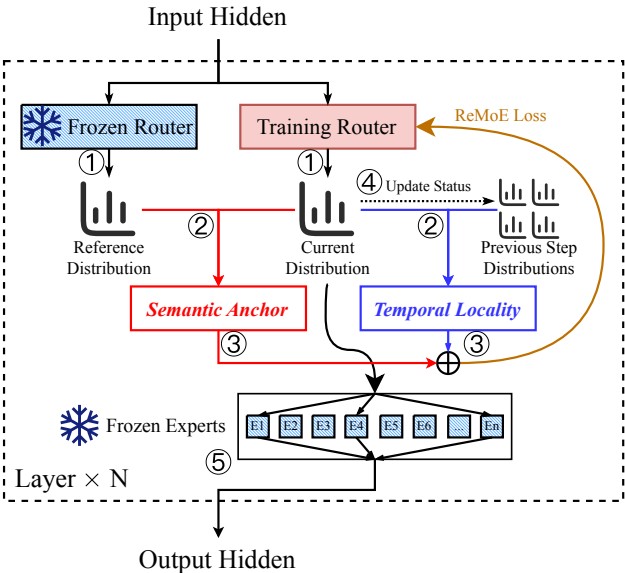

*Figure 3.* **Overview of ReMoE (per MoE layer).** A frozen snapshot gate provides $P_t^{\text{ref}}$ to anchor semantics, while a trainable gate is optimized with temporal-locality regularization using a short routing history buffer; only gate parameters are updated.

cause it explicitly encourages expert dispersion, which conflicts with cache locality under memory constraints.

Our full objective is:

$$\mathcal{L} = L_{\text{CE}} + \lambda_{\text{KL}} L_{\text{Trust}} + \alpha_t L_{\text{Loc}}, \qquad (3)$$

where $L_{\text{Trust}}$ is a semantic anchor (Sec. 4.3), $L_{\text{Loc}}$ is the temporal locality loss (Sec. 4.4), $\lambda_{\text{KL}}$ controls the strength of the anchor, and $\alpha_t \in [0, 1]$ linearly warms up locality regularization during early training (e.g., $\alpha_t = \min(1, t/T_{\text{warm}})$ for training step $t$ and warmup length $T_{\text{warm}}$).

### 4.3. Semantic Anchor

The semantic anchor prevents the router from drifting in a way that harms model quality. At token position $t$, the frozen router applied to the current hidden state produces a reference routing distribution $P_t^{\text{ref}}$ (Figure 3, Step **1**).

**Trust-KL loss.** We anchor the trainable routing distribution $P_t$ to $P_t^{\text{ref}}$ using the Kullback–Leibler (KL) divergence, which measures the discrepancy between two probability distributions. For distributions $P$ and $Q$ over $N_r$ experts,

$$D_{\text{KL}}(P\|Q) = \sum_{k=1}^{N_r} P^{(k)} \log \frac{P^{(k)}}{Q^{(k)}}. \qquad (4)$$

We define

$$L_{\text{Trust}} = \frac{1}{T} \sum_{t=1}^{T} D_{\text{KL}}\Big(P_t \,\|\, \texttt{stop\_gradient}(P_t^{\text{ref}})\Big). \qquad (5)$$

Here $P_t^{\text{ref}}$ is treated as a fixed reference (no gradients flow through the snapshot branch). KL is a natural fit because routing is probabilistic: it directly penalizes distributional drift, places more weight on changes to high-probability experts (which dominate Top-$K$ decisions), and is commonly used as a soft trust-region in distillation and policy optimization (Hinton et al., 2015; Schulman et al., 2017). Anchoring at the distribution level rather than at hidden states applies the constraint exactly at the decision boundary; Appendix C interprets Trust-KL as a soft trust region and gives conditions under which Top-$K$ stability holds under bounded drift.

**Robustness and scope.** Because the anchor is evaluated on the current $h_t$ at every step, $P_t^{\text{ref}}$ adapts to context shifts and the locality bias does not override expert switches required by abrupt semantic transitions—consistent with the Trust-KL ablation in Sec. 5.8, where removing the anchor improves reuse but degrades language-modeling quality. By the same property, on out-of-distribution domains the expected effect is attenuation of the cache-efficiency gain rather than quality degradation, since expert weights remain frozen and Trust-KL bounds drift from the pretrained router. ReMoE is therefore best understood as a deployment-aware routing objective rather than a domain adaptation method.

**Architecture-agnostic design.** ReMoE operates at the distribution level: it uses router outputs $P_t$ (or scores that can be normalized into a distribution) and the selected indices

$E_t$. It is agnostic to the gate parameterization and expert implementation, since experts are never updated. As long as a model exposes token-wise routing outputs and a selection operator (Top-$K$, Top-1/Switch, etc.), the same locality-and-trust shaping objective applies.

## 4.4. Temporal Locality Regularization

Temporal locality regularization reduces token-level routing changes so that the router reuses experts more often. We define the locality loss as a weighted sum:

$$L_{\text{Loc}} = \lambda_{\text{Reuse}} L_{\text{Reuse}} + \lambda_{\text{Smooth}} L_{\text{Smooth}} \\ + \lambda_{\text{Lag}} L_{\text{Lag}} + \lambda_{\text{WS}} L_{\text{WS}}. \tag{6}$$

Here, $L_{\text{Reuse}}$ directly increases short-horizon reuse; $L_{\text{Smooth}}$ suppresses high-frequency jitter; $L_{\text{Lag}}$ suppresses slow drift across longer horizons; and $L_{\text{WS}}$ encourages a compact local working set, which is important for small caches.

**Reuse loss $L_{\text{Reuse}}$.** We encourage $P_t$ to place mass on experts selected at the previous step. Using the reuse mass $m_t$ in Eq. (2), we aggregate reuse at the sequence level:

$$\rho = \frac{1}{T-1} \sum_{t=2}^{T} m_t, \quad L_{\text{Reuse}} = -\log(\rho + 10^{-8}). \tag{7}$$

The small constant $10^{-8}$ is added for numerical stability, preventing $\log(0)$ when $\rho$ is very small and avoiding excessively large gradients early in training. This form increases overall reuse while still allowing occasional necessary switches (since it does not force every step to reuse).

**Smoothness loss $L_{\text{Smooth}}$.** Top-$K$ routing can change abruptly when several experts have similar scores. To reduce such step-to-step jitter, we encourage the routing distribution to change smoothly between adjacent steps. We use the symmetric KL divergence (a symmetric measure of distributional change):

$$\text{SymKL}(P, Q) = \frac{1}{2}\Big(D_{\text{KL}}(P\|Q) + D_{\text{KL}}(Q\|P)\Big), \tag{8}$$

and penalize adjacent changes:

$$L_{\text{Smooth}} = \frac{1}{T-1} \sum_{t=2}^{T} \text{SymKL}(P_t, P_{t-1}). \tag{9}$$

If $P_t$ moves less from one token to the next, Top-$K$ boundaries are crossed less often, improving short-horizon overlap. Unlike the reuse term, which uses the previous routed set as a fixed target, the smoothness term is a purely geometric regularizer on the routing trajectory. It penalizes discrepancies between consecutive distributions, so it must compare $P_t$ and $P_{t-1}$ directly. We do not apply `stop_gradient` here because we want the penalty to propagate to both steps:

$P_t$ should be close to $P_{t-1}$ and, symmetrically, $P_{t-1}$ should be close to $P_t$. This bidirectional coupling yields a stable temporal smoothing effect along the whole sequence, rather than fitting the current step to a fixed past target.

**Lagged inertia loss $L_{\text{Lag}}$.** Smoothness only compares adjacent steps and may miss slow drift: $P_t$ can change slightly each step but still migrate to different experts over many tokens. To suppress this, we align $P_t$ with several earlier distributions using a small lag set $\mathcal{D}$ (e.g., $\{1, 2, 4, 8, 16\}$):

$$L_{\text{Lag}} = \frac{1}{T-1} \sum_{t=2}^{T} \frac{1}{|\mathcal{D}|} \sum_{\substack{d \in \mathcal{D} \\ t-d \geq 1}} \text{SymKL}(P_t, P_{t-d}). \tag{10}$$

The factor $1/|\mathcal{D}|$ normalizes the loss across lags. While $L_{\text{Smooth}}$ suppresses adjacent-step jitter, $L_{\text{Lag}}$ curbs slower multi-step drift.

**Working-set compression loss $L_{\text{WS}}$.** Reuse and smoothness alone do not prevent the router from gradually visiting many experts over a longer span. We therefore encourage routing to concentrate within local windows. For a window size $W$, we average distributions within each window:

$$\bar{P}_b = \frac{1}{W} \sum_{j=1}^{W} P_{(b-1)W+j}, \quad b = 1, \ldots, n, \quad n = \lfloor T/W \rfloor. \tag{11}$$

We then minimize the entropy of the window-averaged distribution, where $H(P) = -\sum_{k=1}^{N_r} P^{(k)} \log P^{(k)}$ measures how spread a distribution is (smaller means more concentrated):

$$L_{\text{WS}} = \frac{1}{n} \sum_{b=1}^{n} H(\bar{P}_b). \tag{12}$$

This encourages each local window to rely on a smaller subset of experts, aligning routing with small cache capacities, while the Trust-KL term limits pathological collapse.

**Routing imbalance.** The locality regularizers, especially $L_{\text{Reuse}}$ and $L_{\text{WS}}$, can increase the load-balance coefficient of variation (CV) by concentrating routing decisions. This trade-off is acceptable in our target setting ($B=1$, no expert parallelism), where local concentration reduces distinct expert loads; the Trust-KL anchor limits excessive collapse.

## 5. Evaluation

We evaluate ReMoE along four dimensions: (i) routing locality and inference-time expert balance; (ii) trace-driven cache efficiency under standard replacement policies; (iii) real-system serving latency under expert offloading; and (iv) capability preservation and attribution against generic router continued fine-tuning.

## 5.1. Experimental Setup

**Models.** Unless otherwise specified, we use DeepSeek-V2-Lite, a fine-grained MoE LLM with 15.7B total parameters and 2.4B activated parameters per token. The model has 27 Transformer layers, of which 26 are MoE layers after the first dense layer. Each MoE layer contains 64 routed experts plus 2 shared experts and uses Top-$K$=6 routing.

**Data and preprocessing.** We fine-tune on OpenHermes-2.5 (Teknium, 2023), a multi-turn instruction/chat corpus spanning general chat, reasoning, code, and math. We serialize each example into a role-prefixed transcript ("User:", "Assistant:") and append end-of-sequence (EOS) token. We use 100,000 samples for training and 1,000 held-out samples for evaluation.

**Fine-tuning setup.** We fine-tune for 2,000 steps with AdamW (learning rate $5 \times 10^{-5}$, linear warmup 200 steps), BF16, gradient clipping 1.0, and sequence length 2048. We use micro-batch size 1 with gradient accumulation 8.

**Loss weights and schedules.** We use the full ReMoE objective with a warmup schedule for locality regularization; all hyperparameters (including $\mathcal{D}$ and $W$) follow Appendix E.

**Baselines and ablations.** We compare against the pre-trained router (Baseline) and a cross-entropy-only (CE-only) router fine-tuning baseline. CE-only uses the same data, optimizer, training length, and frozen-parameter setting as ReMoE, but optimizes only $L_{\text{CE}}$, i.e., the standard next-token cross-entropy loss, without the temporal-locality regularizers or Trust-KL anchor; this isolates the effect of the locality objective from generic continued router adaptation. For ablations, we remove one component at a time while keeping the recipe fixed: w/o Trust ($\lambda_{\text{KL}}$=0), w/o Reuse ($\lambda_{\text{reuse}}$=0), and w/o Consistency ($\lambda_{\text{smooth}}$=$\lambda_{\text{lag}}$=$\lambda_{\text{ws}}$=0).

**Real-system serving setup.** We evaluate ReMoE under two serving-side expert-offloading settings. First, we use vLLM (Kwon et al., 2023) with the MoE expert-offloading implementation (vLLM Contributors, 2026) on a 24GB GPU connected through a PCIe Gen3 ×16 host-device link. We set `max-num-seqs=1`, `moe-expert-cache-size=6`, disable prefix caching and chunked prefill, and evaluate with concurrency 1. Second, we evaluate an edge-oriented setup on Jetson Orin NX 16GB using `llama.cpp` (ggml-org, 2026). The model is stored on an aigo NVMe SSD DP35 256GB and connected through PCIe Gen3 ×4.

**Expert-cache simulation.** For routing-locality and cache-efficiency evaluation, we focus on $B$=1 autoregressive decoding and record the Top-$K$ expert indices at each decode step. EOR is computed directly from these routing traces. To estimate cache pressure independently of a specific serving implementation, we run a per-layer expert-cache simula-

*Table 2.* **Routing metrics under teacher forcing** ($B$=1)**.** Rel. $\Delta$ is (ReMoE − Baseline)/Baseline.

| Method | EOR ↑ | Entropy ↓ | CV ↑ |
|--------|-------|-----------|------|
| Baseline | 27.3% | 0.9998 | 0.0409 |
| CE-only | 22.9% | 0.9998 | 0.0392 |
| ReMoE | 34.5% | 0.9971 | 0.1608 |
| Rel. $\Delta$ | +26.4% | −0.27% | +293.1% |

tor with capacity $C$ (experts/layer) under standard replacement policies, using the recorded routing traces as input. At each decode step, the Top-$K$ experts form one request: resident experts count as hits, while non-resident experts count as loads and may trigger evictions.

## 5.2. Routing Locality and Inference-Time Expert Imbalance

**Trajectory.** Figure 2 visualizes Top-$K$ expert activations over decoding steps. Compared to the baseline, ReMoE produces longer contiguous expert streaks and fewer abrupt switches, suggesting reduced routing jitter.

**Metrics and trade-off.** Table 2 quantifies this effect. ReMoE increases EOR from 27.3% to 34.5% ($\Delta$=+7.2 points). Routing becomes moderately more concentrated: entropy decreases from 0.9998 to 0.9971 ($\Delta$=−0.0027), and the load-balance CV increases from 0.0409 to 0.1608 ($\Delta$=+0.1199). This pattern matches the design of ReMoE: locality regularizers encourage reuse, while the trust objective constrains the router from drifting too far from the pretrained routing distribution. Sequence-level expert diversity is preserved: unique experts visited per sequence changes negligibly (64.000 → 63.997), confirming that the concentration is step-level rather than a global routing collapse.

## 5.3. Cache Efficiency

We use a trace-driven per-layer expert cache simulator with request-level resets, i.e., each prompt starts from an empty expert cache. We sample 128 prompts from `ShareGPT_V3_unfiltered`, run greedy decoding with $B$=1 and `max_new_tokens`=64, and record the Top-$K$ expert indices at each decode step. At each layer-step, we deduplicate repeated expert indices within the Top-$K$ routing result and treat the resulting distinct expert set as one cache request. We report unique hit rate (uHR) and total unique misses (#uMiss), aggregated over all MoE layers and prompts.

Table 3 shows that ReMoE consistently reduces expert loads under LRU. At the Top-$K$-matched setting $C$=6, uHR improves from 0.3187 to 0.3687, while #uMiss drops from 0.8707M to 0.8068M. Similar improvements under LFU/FIFO and Belady's optimal policy, as well as the

*Table 3.* **Expert cache efficiency under LRU.** $\Delta$ is ReMoE−Baseline. #uMiss is reported in millions. Full LFU/FIFO, Belady, and TPOT-proxy results are in Appendix F.1.

| $C$ | uHR | uHR-R | $\Delta$uHR | #uMiss | #uMiss-R | $\Delta$#uMiss |
|---|---|---|---|---|---|---|
| 4 | 0.2058 | 0.2374 | +0.0316 | 1.0150 | 0.9746 | -0.0404 |
| 6 | 0.3187 | 0.3687 | +0.0500 | 0.8707 | 0.8068 | -0.0639 |
| 8 | 0.3629 | 0.4142 | +0.0513 | 0.8141 | 0.7486 | -0.0655 |
| 12 | 0.4519 | 0.5035 | +0.0516 | 0.7005 | 0.6345 | -0.0660 |

*Table 4.* Evaluation with vLLM expert offloading (RTX 3090, `moe-expert-cache-size=6`, ShareGPT prompts).

| Method | Tok/s ↑ | TTFT (ms) ↓ | TPOT (ms) ↓ | uHR ↑ |
|---|---|---|---|---|
| Baseline | 3.58 | 769.23 | 254.31 | 39.4% |
| CE-only | 2.95 | 780.12 | 286.82 | 21.1% |
| ReMoE | 3.88 | 758.27 | 242.99 | 43.4% |
| vs. Baseline | +8.4% | −1.4% | −4.5% | +3.9 pp |

corresponding step-level TPOT proxy, are reported in Appendix F.1.

### 5.4. Real-System Serving Evaluation

**vLLM expert offloading.** Table 4 reports serving metrics alongside CE-only for attribution. ReMoE improves output throughput from 3.58 to 3.88 tok/s, reduces mean TPOT from 254.31 ms to 242.99 ms, and raises the average per-layer unique-expert hit rate from 39.4% to 43.3%. CE-only is substantially worse than both the pretrained baseline and ReMoE, indicating that the serving gain is not explained by generic router continued fine-tuning.

**Edge evaluation on Jetson Orin NX.** Table 5 reports results on Jetson Orin NX 16GB with `llama.cpp`, where the model is stored on NVMe SSD and experts are served through a slower storage path under a tighter memory budget. ReMoE consistently reduces both TTFT and TPOT across all three workload types.

**Analysis.** The vLLM result provides conservative server-side validation: the PCIe host-device path partially hides expert-cache miss cost, limiting the observable gain. The Jetson result better represents our target setting: with slower SSD-backed expert storage, cache misses are more expensive and ReMoE's reduction in unique expert loads (Sec. 5.3) translates into 43.6–49.8% steady-state TPOT reduction across workloads.

### 5.5. Capability Preservation

We evaluate downstream tasks with `lm-eval-harness` (lm_eval) (Gao et al., 2024), reporting GSM8K exact match (strict and flexible) (Cobbe et al., 2021), HumanEval pass@1 (Chen et al., 2021), MMLU (Hendrycks et al., 2021), and IFEval (Zhou et al., 2023) accuracy. Baseline

*Table 5.* Edge evaluation on Jetson Orin NX 16GB + `llama.cpp` (`-np 1, -n 128, --mmap`). All latencies are in ms. $\Delta$ denotes the relative change from Baseline to ReMoE, computed as $(\text{ReMoE} - \text{Base})/\text{Base}$; negative values indicate latency reduction. Decode speed is $\text{TPOT}_{\text{Base}}$ / $\text{TPOT}_{\text{ReMoE}}$.

| Workload | TTFT (ms) ↓ | | | TPOT (ms) ↓ | | | Decode |
|---|---|---|---|---|---|---|---|
| | Base | ReMoE | $\Delta$ | Base | ReMoE | $\Delta$ | Speed |
| ShareGPT | 7150.12 | 5368.77 | −24.9% | 554.69 | 306.27 | −44.8% | 1.81× |
| GSM8K | 6041.65 | 4736.70 | −21.6% | 613.73 | 346.04 | −43.6% | 1.77× |
| HumanEval | 7185.78 | 5233.11 | −27.2% | 672.68 | 337.61 | −49.8% | 1.99× |

*Table 6.* **Capability on standard benchmarks (lm-eval).** $\Delta$ is ReMoE−Baseline in percentage points.

| Benchmark (metric) | Baseline | CE-only | ReMoE | $\Delta$ (pp) |
|---|---|---|---|---|
| GSM8K (EM, strict) | 38.89 | 36.92 | 38.13 | −0.76 |
| GSM8K (EM, flex) | 39.04 | 37.23 | 38.36 | −0.68 |
| HumanEval (pass@1) | 26.83 | 28.05 | 29.27 | +2.44 |
| MMLU (acc) | 57.72 | 57.44 | 57.81 | +0.09 |
| IFEval (prompt loose) | 17.93 | — | 17.93 | 0.00 |
| IFEval (prompt strict) | 17.19 | — | 16.08 | −1.11 |

and ReMoE use the same backend and task configurations.

**Results.** Table 6 shows that ReMoE preserves overall capability. MMLU remains essentially unchanged ($\Delta$=+0.09 pp). HumanEval improves from 26.83% to 29.27%. GSM8K slightly decreases (strict: $\Delta$=−0.76 pp; flexible: $\Delta$=−0.68 pp), within the reported uncertainty. Overall, we do not observe evidence that the locality gains in Sec. 5.2–5.3 come at the cost of benchmark performance.

### 5.6. Attribution: Locality Objective vs. Generic Router Adaptation

A possible alternative explanation for ReMoE's gains is that any continued fine-tuning of the router on OpenHermes-2.5 would produce similar improvements. The CE-only column in Table 6 rules this out: CE-only router fine-tuning, which uses identical training conditions without the locality objective or Trust-KL anchor, reduces EOR to 0.2293—below the pretrained baseline (0.2730)—and degrades GSM8K scores relative to the baseline without improving MMLU or HumanEval. The serving results in Table 4 show the same pattern: CE-only worsens throughput and TPOT to below-baseline levels. The locality gain therefore requires the explicit locality-aware objective; generic router adaptation on the fine-tuning distribution is not sufficient.

### 5.7. Generalization across LLMs and GPUs/NPUs

To test model-level generalization beyond DeepSeek, we apply the same gate-only recipe to Qwen1.5-MoE-A2.7B without tuning hyperparameters. ReMoE increases short-horizon overlap (EOR: $0.1695 \rightarrow 0.2156$; +27.2% relative) while moderately concentrating routing (entropy: 0.99996

$\rightarrow$ 0.99861; CV: 0.0174 $\rightarrow$ 0.1109). Downstream capability remains comparable on `lm-eval` (Appendix F.4).

We further test whether gate-only fine-tuning followed by serving evaluation can be executed on an Iluvatar GPU platform. Specifically, we first perform gate-only fine-tuning for DeepSeek-V2-Lite, and then evaluate the resulting model with `expert-kit` (Expert Kit Contributors, 2026) on a server with two BI-V150 GPUs using prompts sampled from ShareGPT. The expert cache size in `expert-kit` serving is set to 800. The storage device is a NVMe SSD connected through PCIe 3.0 $\times$4. After 2,000 fine-tuning steps, ReMoE improves EOR from 0.2763 to 0.3454 (+25.0% relative). In serving, ReMoE improves prefill throughput from 0.94 to 2.32 tokens/s, corresponding to a 2.47$\times$ prefill speedup. For decoding, ReMoE improves decode throughput from 1.08 to 1.66 tokens/s, corresponding to a 1.54$\times$ decode speedup. Overall, ReMoE achieves a 1.67$\times$ end-to-end inference speedup.

We also run gate-only fine-tuning and generation evaluation on a Kunpeng–Ascend platform. Specifically, we apply ReMoE to Qwen1.5-MoE-A2.7B on a Kunpeng-920 server with a single Ascend 910B3 NPU, including gate adaptation and `llama.cpp` generation evaluation on prompts sampled from ShareGPT. The model is stored on a Huawei HWE6AP443T8L00KN NVMe SSD with a PCIe Gen4 $\times$4 connection. After 2,000 fine-tuning steps, ReMoE improves EOR from 0.1672 to 0.2178 (+30.2% relative), and improves generation throughput from 4.3 to 4.8 tokens/s (+11.6%). Together with the Iluvatar results, this provides preliminary evidence that ReMoE transfers across MoE model families, expert-offloading runtimes, and heterogeneous accelerator platforms.

### 5.8. Ablation Studies

The CE-only control in Sec. 5.6 isolates the full ReMoE objective from generic router continued fine-tuning. We now ablate the internal components of the ReMoE objective, removing one term at a time while keeping all other settings fixed; results are summarized in Table 7.

**Reuse drives locality.** Removing Reuse largely eliminates the locality gain (EOR: 0.345 $\rightarrow$ 0.283; $\Delta$=$-$0.062), and the router shifts back toward near-uniform behavior (entropy: 0.9971 $\rightarrow$ 0.9997; $\Delta$=+0.0026; CV: 0.1608 $\rightarrow$ 0.0536; $\Delta$=$-$0.1072). This shows that overlap improvements are not a byproduct of other regularizers.

**Consistency terms stabilize trajectories.** Disabling Smooth/Lag/WS yields a smaller but consistent drop in EOR (0.345 $\rightarrow$ 0.329; $\Delta$=$-$0.016), suggesting these terms primarily reduce boundary-crossing jitter and slow drift rather than redefining the global routing distribution.

**Trust prevents over-concentration.** Without Trust,

*Table 7.* **Ablation results.** $\Delta$ is w.r.t. the full ReMoE objective. PPL denotes perplexity. Acc@1/Acc@5 report token-level next-token prediction accuracy, where the ground-truth next token must appear in the model's top-1/top-5 predicted tokens.

| Method | PPL↓ | Acc@1↑ | Acc@5↑ | EOR↑ | Entropy↓ | CV↑ |
|---|---|---|---|---|---|---|
| Ours (Full) | 3.2280 | 71.78 | 89.65 | 0.3453 | 0.9971 | 0.1608 |
| w/o Consistency | 3.2254 | 71.81 | 89.64 | 0.3290 | 0.9977 | 0.1436 |
| w/o Reuse | 3.2222 | 71.81 | 89.65 | 0.2831 | 0.9997 | 0.0536 |
| w/o Trust | 3.2629 | 71.58 | 89.54 | 0.3877 | 0.9950 | 0.2110 |
| $\Delta$ (w/o Cons.) | $-0.0026$ | $+0.03$ | $-0.01$ | $-0.0163$ | $+0.0006$ | $-0.0172$ |
| $\Delta$ (w/o Reuse) | $-0.0058$ | $+0.03$ | $+0.00$ | $-0.0622$ | $+0.0026$ | $-0.1072$ |
| $\Delta$ (w/o Trust) | $+0.0349$ | $-0.20$ | $-0.11$ | $+0.0424$ | $-0.0021$ | $+0.0502$ |

EOR becomes the highest (0.388; $\Delta$=+0.043 vs. full), but routing becomes more concentrated (entropy: 0.9971 $\rightarrow$ 0.9950; $\Delta$=$-$0.0021; CV: 0.1608 $\rightarrow$ 0.2110; $\Delta$=+0.0502), accompanied by slightly worse language modeling (PPL: 3.2280 $\rightarrow$ 3.2629; $\Delta$=+0.0349; Acc@1: 71.78 $\rightarrow$ 71.58; $\Delta$=$-$0.20). This supports the role of the frozen-reference anchor in preserving routing semantics while allowing locality improvements.

## 6. Conclusion

We propose ReMoE, a router-only fine-tuning method that improves short-horizon expert reuse for memory-constrained MoE inference without changing the model architecture or inference graph. Across DeepSeek and Qwen MoE models, ReMoE improves routing locality, cache friendliness, and real-system decoding efficiency while largely preserving downstream capability.

## Impact Statement

This paper presents research intended to advance the field of machine learning. Although the work may have broader societal implications, we do not identify any specific societal consequences that require emphasis in this submission.

## Acknowledgements

This work is supported by the National Natural Science Foundation of China (Grant No. 62572022), National Key R&D Program of China (Grant No. 2023YFB4503100), HUAWEI (TC20250908049), BUAA Kunpeng&Ascend Center of Cultivation, the Fundamental Research Funds for the Central Universities, and Guangdong S&T Program (2025B0101080001).

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

# A. Connecting EOR to Expert Loads

This appendix formalizes why the Expert Overlap Ratio (EOR) is a useful proxy for expert-weight I/O under memory-constrained MoE decoding. The key message is that, under a standard per-layer expert cache model, step-to-step expert overlap controls an upper bound on the number of experts that must be fetched from external storage.

## A.1. Setup and Cache Model

We consider a single MoE layer under autoregressive decoding with routed Top-$K$ expert requests $\{E_t\}_{t=1}^T$, where $E_t \subseteq \{1, \ldots, N_r\}$ and $|E_t| = K$ for each step $t$. (Shared experts, if any, are excluded from $E_t$ and are treated as always-resident; cf. Sec. 3.1.)

**Cache state.** Let $\mathcal{C}_t$ denote the set of experts resident in the per-layer expert cache at the beginning of step $t$ (i.e., before serving the request $E_t$), with capacity

$$|\mathcal{C}_t| \leq C. \tag{13}$$

**Expert fetches (cache misses).** Define the number of experts that must be fetched from external storage at step $t$ as

$$N_{\text{fetch}}(t) = |E_t \setminus \mathcal{C}_t| = K - |E_t \cap \mathcal{C}_t|. \tag{14}$$

**Serve-and-admit cache update.** We assume a standard cache update semantics: after serving step $t$ (executing the experts in $E_t$), the cache is updated according to a replacement policy to produce the next-step state $\mathcal{C}_{t+1}$. The only property we will need is that the cache admits the requested experts when capacity permits.

## A.2. Assumptions and Main Claim

We spell out the assumptions under which EOR yields a deterministic fetch bound.

**Assumption A.1** (Capacity). $C \geq K$, i.e., the cache can hold the $K$ experts required by a single decoding step.

**Assumption A.2** (Step atomicity and admission). Within a decoding step, expert weights that are fetched/used for the request $E_t$ are not evicted before the step completes. Moreover, after serving step $t$, the cache update produces a state $\mathcal{C}_{t+1}$ that contains the just-served experts:

$$E_t \subseteq \mathcal{C}_{t+1}. \tag{15}$$

**Assumption A.3** (Cache isolation (no interference)). The per-layer expert cache is isolated from other memory traffic (KV cache growth, activations, non-expert weights, other layers, or other requests), so that such traffic does not insert into / evict from the expert cache between steps.

Assumption A.2 matches typical expert-caching implementations: once an expert is needed for the current token, its weights are loaded (if missing), used, and then retained (unless capacity forces evicting older experts). Common policies such as LRU/LFU/FIFO satisfy (15) under $C \geq K$ by evicting experts not in $E_t$.

**Instantaneous reuse and EOR.** Recall the step-to-step overlap metrics from Sec. 3.2:

$$\text{IR}_t = \frac{|E_t \cap E_{t-1}|}{K}, \qquad \text{EOR} = \frac{1}{T-1} \sum_{t=2}^T \text{IR}_t. \tag{16}$$

**Proposition A.4** (EOR upper-bounds expert fetches (per step and on average)). *Under Assumptions A.1–A.3, for all $t \geq 2$,*

$$N_{\text{fetch}}(t) \leq K - |E_t \cap E_{t-1}| = K(1 - \text{IR}_t). \tag{17}$$

*Consequently, the average fetches over a length-$T$ trajectory satisfy*

$$\bar{N}_{\text{fetch}} = \frac{1}{T-1} \sum_{t=2}^T N_{\text{fetch}}(t) \leq K(1 - \text{EOR}). \tag{18}$$

## A.3. Proof

The proof reduces to showing that experts used at step $t-1$ must be present at the beginning of step $t$.

**Lemma A.5** (Previous-step experts remain resident). *Under Assumptions A.1–A.3, we have*

$$E_{t-1} \subseteq \mathcal{C}_t \qquad \text{for all } t \geq 2. \tag{19}$$

*Proof.* By Assumption A.2, after serving step $t-1$ the cache state at the next step satisfies $E_{t-1} \subseteq \mathcal{C}_t$ (this is exactly (15) with $t \leftarrow t-1$). Assumption A.3 rules out any inter-step interference that could evict these experts before step $t$ begins. $\square$

*Proof of Proposition A.4.* By definition (14),

$$N_{\text{fetch}}(t) = K - |E_t \cap \mathcal{C}_t|. \tag{20}$$

From Lemma A.5, $E_{t-1} \subseteq \mathcal{C}_t$ for all $t \geq 2$. Intersecting both sides with $E_t$ yields

$$E_t \cap E_{t-1} \subseteq E_t \cap \mathcal{C}_t \quad \implies \quad |E_t \cap \mathcal{C}_t| \geq |E_t \cap E_{t-1}|. \tag{21}$$

Substituting into the fetch definition gives the per-step bound

$$N_{\text{fetch}}(t) \leq K - |E_t \cap E_{t-1}| = K(1 - \text{IR}_t), \tag{22}$$

which is (17). Averaging over $t = 2, \ldots, T$ immediately yields (18):

$$\bar{N}_{\text{fetch}} = \frac{1}{T-1} \sum_{t=2}^{T} N_{\text{fetch}}(t) \leq \frac{1}{T-1} \sum_{t=2}^{T} K(1 - \text{IR}_t)$$

$$= K \left( 1 - \frac{1}{T-1} \sum_{t=2}^{T} \text{IR}_t \right) = K(1 - \text{EOR}). \tag{23}$$

$\square$

**Tightness (when the bound is achieved).** The bound (17) can be tight. For example, when $C = K$ and the cache contains exactly $E_{t-1}$ at the beginning of step $t$, we have $|E_t \cap \mathcal{C}_t| = |E_t \cap E_{t-1}|$ and therefore $N_{\text{fetch}}(t) = K - |E_t \cap E_{t-1}|$.

## A.4. When the EOR Bound Can Fail (Assumptions and Counterexamples)

Proposition A.4 is deliberately stated for a clean per-layer cache model. Below we list concrete failure modes (counterexamples) showing why each assumption matters:

1. **If $C < K$, misses are unavoidable.** When capacity is insufficient to hold a full routed set, then regardless of routing overlap, at least $K - C$ experts must be missing at every step. For instance, if $K = 6$ and $C = 4$, even if $E_t = E_{t-1}$, the cache cannot hold all 6 experts simultaneously, so a guarantee of the form (17) no longer holds. This is why we restrict to $C \geq K$.

2. **If cache isolation is violated, $E_{t-1} \subseteq \mathcal{C}_t$ may fail.** Suppose expert weights share the same memory pool with other objects that can evict experts between steps (e.g., KV-cache expansion, activation buffers, or a different layer/request). Then even if $E_{t-1}$ was fully loaded during step $t-1$, some of these experts might be evicted before step $t$ begins, and the containment in Lemma A.5 can fail. In that case, $|E_t \cap \mathcal{C}_t|$ may be smaller than $|E_t \cap E_{t-1}|$.

3. **Inter-step insertions (e.g., aggressive prefetch) can break the guarantee.** If a prefetcher inserts additional experts into the cache between step $t-1$ and $t$ and triggers evictions, it may evict members of $E_{t-1}$ even under $C \geq K$. For example, with $C = K$, any insertion of an expert not in $E_{t-1}$ forces an eviction; if the policy/prefetcher evicts from $E_{t-1}$, then $E_{t-1} \nsubseteq \mathcal{C}_t$. The EOR bound is therefore best interpreted as a router-intrinsic guarantee under a cache model without inter-step insertions.

4. **Non-admitting or unusual policies.** Assumption A.2 requires that the cache admits the experts it just served (when $C \geq K$). Pathological policies that can evict an expert immediately after it is fetched/used (within the same step), or that refuse to retain the requested set, can violate (15) and invalidate the proof. Such policies are atypical for expert-weight caching but are included here for completeness.

## A.5. Generalizations Beyond Immediate Overlap

Proposition A.4 uses immediate step-to-step reuse because it is the part that can be guaranteed for any cache with $C \geq K$ under the above step semantics. When $C$ is larger, the cache can preserve experts from a longer recent history, yielding a stronger bound.

**Longer-horizon working-set bound (LRU-style insight).** Define the distinct expert working set over the last $\ell$ steps:

$$U_{t,\ell} = \bigcup_{j=1}^{\ell} E_{t-j}. \tag{24}$$

Let

$$L_t = \max\{\ell \geq 1 : |U_{t,\ell}| \leq C\}. \tag{25}$$

If the cache update policy preserves the most recently used distinct experts (as LRU does under cache isolation), then the experts in $U_{t,L_t}$ remain resident at the beginning of step $t$, i.e., $U_{t,L_t} \subseteq C_t$. Therefore,

$$N_{\text{fetch}}(t) = K - |E_t \cap C_t| \leq K - |E_t \cap U_{t,L_t}|. \tag{26}$$

This highlights that larger cache capacity $C$ can exploit longer-horizon reuse beyond $E_{t-1}$. Our EOR-based bound corresponds to the always-valid case $\ell = 1$, since $|U_{t,1}| = |E_{t-1}| = K \leq C$.

**Generality across replacement policies.** The immediate-overlap bound in Proposition A.4 does not require a specific policy such as LRU. It only relies on Assumption A.2 (the just-served set is retained) and isolation. Thus, the per-step EOR bound applies to common policies used in our simulator (LRU/LFU/FIFO) as long as they satisfy the serve–admit semantics and do not perform inter-step evictions of $E_{t-1}$. In contrast, the longer-horizon bound (26) is most naturally justified for LRU-like policies that explicitly preserve recently used distinct experts.

**Extension to multiple MoE layers.** The above analysis is per-layer. If a model contains $L_{\text{moe}}$ MoE layers and each layer maintains an independent cache of capacity $C$, then the total number of expert fetches at step $t$ is

$$N_{\text{fetch}}^{\text{total}}(t) = \sum_{\ell=1}^{L_{\text{moe}}} N_{\text{fetch}}^{(\ell)}(t), \tag{27}$$

and Proposition A.4 applies to each layer separately, yielding an immediate bound on the total by summation.

**Summary.** EOR provides an always-valid short-horizon cacheability proxy under $C \geq K$ with standard serve–admit cache semantics. Larger caches and LRU-like policies can additionally benefit from longer-horizon reuse, for which (26) motivates windowed/working-set viewpoints consistent with our locality regularizers in Sec. 4.4.

## B. Why Reuse Mass is a Valid Surrogate

ReMoE optimizes a differentiable surrogate for step-to-step Top-$K$ overlap. This section provides a simple justification that the reuse mass

$$m_t = \frac{1}{K} \sum_{k \in \tilde{E}_{t-1}} P_t^{(k)} \qquad (\text{Eq. }(2)) \tag{28}$$

is aligned with expected overlap under an analysis-only stochastic routing rule.

**Analysis-only stochastic Top-$K$.** Consider a stochastic routing mechanism that samples $K$ expert indices $X_{t,1}, \ldots, X_{t,K}$ i.i.d. from the categorical distribution $P_t$. Let $\tilde{E}_{t-1}$ be the previous-step routed set treated as fixed (i.e., stop_gradient as in Sec. 4). Define the random variable

$$R_t = \sum_{j=1}^{K} \mathbf{1}\left\{X_{t,j} \in \tilde{E}_{t-1}\right\}, \tag{29}$$

where $\mathbf{1}\{\cdot\}$ is the indicator function that equals 1 if the condition holds and 0 otherwise. The random variable counts (with multiplicity) how many of the $K$ sampled experts belong to the previous set.

**Lemma B.1** (Reuse mass equals expected reused samples). *Under the i.i.d. sampling rule above,*

$$\mathbb{E}[R_t \mid \tilde{E}_{-1}] = K \sum_{k \in \tilde{E}_{t-1}} P_t^{(k)} = K^2 \, m_t. \tag{30}$$

*Proof.* By linearity of expectation and i.i.d. sampling,

$$\mathbb{E}[R_t \mid \tilde{E}_{t-1}] = \sum_{j=1}^{K} \Pr(X_{t,j} \in \tilde{E}_{t-1}) = K \sum_{k \in \tilde{E}_{t-1}} \Pr(X_{t,1} = k)$$

$$= K \sum_{k \in \tilde{E}_{t-1}} P_t^{(k)} = K^2 m_t, \tag{31}$$

which proves (30). $\qquad\square$

**Implication.** Lemma B.1 shows that increasing $m_t$ increases the expected number of reused expert samples under this stochastic routing rule. While deployed MoE routing uses deterministic Top-$K$ (a set of size $K$), $m_t$ remains a natural surrogate because it directly increases the probability mass assigned to experts that were selected at the previous step, thereby making it more likely that those experts stay within the next step's Top-$K$ boundary.

**Set overlap vs. multiplicity.** If one converts the sampled multiset $\{X_{t,j}\}_{j=1}^{K}$ into a unique set, the resulting set overlap $|E_t \cap \tilde{E}_{t-1}|$ is upper-bounded by $R_t$ (duplicates in sampling only increase $R_t$). Thus, although (30) does not equal the expected set overlap exactly, it provides an aligned and differentiable signal.

**Why we stop gradients through $\tilde{E}_{t-1}$.** Treating $\tilde{E}_{t-1}$ as a constant yields a one-way learning signal that matches autoregressive decoding: we encourage the current distribution $P_t$ to reuse the previous realized expert set.

**Remark.** The stochastic rule above is used only to justify the surrogate; ReMoE itself is trained and deployed with the standard deterministic Top-$K$ router.

## C. Trust-KL as a Soft Trust Region

This section provides additional interpretation for the Trust-KL anchor (Eq. (5)) as a soft trust region on routing behavior, and explains when small drift implies stable Top-$K$ expert selection.

Our goal is to constrain routing decisions while leaving the backbone computation and expert weights untouched. Anchoring hidden states (e.g., $\|h_t - h_t^0\|$) would require storing reference trajectories and may over-constrain representations, potentially conflicting with the LM objective. In contrast, anchoring $P_t$ is lightweight and targeted: $P_t^{\text{ref}}$ is computed on-the-fly from the frozen snapshot given the current $h_t$, and the constraint is applied exactly at the decision boundary that determines expert selection. This keeps the regularization architecture-agnostic and limits semantic drift without restricting the rest of the network.

### C.1. From KL to Distributional Stability

Let $P_t$ be the trainable routing distribution and $P_t^{\text{ref}}$ the frozen reference distribution. By Pinsker's inequality, for any step $t$,

$$\|P_t - P_t^{\text{ref}}\|_1 \;\leq\; \sqrt{2\, D_{\mathrm{KL}}(P_t \| P_t^{\text{ref}})}. \tag{32}$$

Therefore, minimizing $L_{\text{Trust}}$ controls a strong notion of distributional drift: small Trust-KL implies $P_t$ stays close to $P_t^{\text{ref}}$ in $L_1$ (total variation up to a factor $1/2$).

### C.2. Top-$K$ Stability Under a Probability Margin

We next give a sufficient condition under which the Top-$K$ set does not change. Unlike a score/logit margin, the condition below is stated directly on the routing distributions, matching our distribution-level anchor.

**Lemma C.1** (Top-$K$ stability under a probability margin). *Let $Q \in \mathbb{R}^{N_r}$ be a reference routing distribution (e.g., $Q = P_t^{\mathrm{ref}}$) and let $q_{(1)} \geq \cdots \geq q_{(N_r)}$ denote its entries sorted in descending order. Define the $K$-boundary probability margin*

$$\gamma = q_{(K)} - q_{(K+1)} \; > \; 0. \tag{33}$$

*If another distribution $P \in \mathbb{R}^{N_r}$ satisfies*

$$\|P - Q\|_\infty \; < \; \gamma/2, \tag{34}$$

*then the Top-$K$ index set is unchanged:*

$$\mathrm{Top\text{-}K}(P) = \mathrm{Top\text{-}K}(Q). \tag{35}$$

*Proof.* Let $S$ be the Top-$K$ index set of $Q$, and let $i^\star \in S$ and $j^\star \notin S$ be such that $Q_{i^\star} = q_{(K)}$ and $Q_{j^\star} = q_{(K+1)}$. For any $i \in S$ and $j \notin S$, we have $Q_i \geq q_{(K)}$ and $Q_j \leq q_{(K+1)}$. Under (34),

$$P_i \; \geq \; Q_i - \gamma/2 \; \geq \; q_{(K)} - \gamma/2, \qquad P_j \; \leq \; Q_j + \gamma/2 \; \leq \; q_{(K+1)} + \gamma/2. \tag{36}$$

Hence,

$$P_i - P_j \; \geq \; (q_{(K)} - \gamma/2) - (q_{(K+1)} + \gamma/2) = \gamma - \gamma = 0, \tag{37}$$

and strict positivity holds because $\|P - Q\|_\infty < \gamma/2$. Therefore no index outside $S$ can overtake an index inside $S$, so the Top-$K$ set is unchanged. $\square$

**Interpretation.** Lemma C.1 formalizes that Top-$K$ selections are robust when there is a nontrivial separation between the $K$-th and $(K+1)$-th experts in the reference distribution. Near tie regions (small $\gamma$), even small shifts can flip membership across the boundary. The Trust-KL anchor reduces distributional drift (Eq. (32)), making large boundary-crossing changes less likely, while still allowing necessary switches when the LM objective and locality regularization favor them.

**Why a soft trust region.** Trust-KL does not impose a hard constraint such as (34). Instead, it penalizes deviations in the output distribution (Eq. (5)), which is lightweight and architecture-agnostic, and empirically sufficient to prevent extreme routing drift.

## D. Extended Notation

*Table 8.* **Extended notation used in training and implementation.**

| Symbol | Meaning |
|---|---|
| $\mathcal{D}$ | lag set for inertia regularization (e.g., $\{1, 2, 4, 8, 16\}$) |
| $W$ | window size for working-set/entropy regularization |
| $\alpha_t$ | locality warmup schedule coefficient at training step $t$ |
| $T_{\mathrm{warm}}$ | warmup length (steps) for locality regularization |
| $\lambda_{\mathrm{Reuse}}$ | weight of reuse loss $L_{\mathrm{Reuse}}$ |
| $\lambda_{\mathrm{Smooth}}$ | weight of smoothness loss $L_{\mathrm{Smooth}}$ |
| $\lambda_{\mathrm{Lag}}$ | weight of lagged inertia loss $L_{\mathrm{Lag}}$ |
| $\lambda_{\mathrm{WS}}$ | weight of working-set compression loss $L_{\mathrm{WS}}$ |
| $\epsilon$ | small constant for numerical stability |

## E. Implementation Details

### E.1. Training Details

**Hardware.** All fine-tuning and trace collection runs are executed on a single node equipped with one CPU (2 sockets, 56 physical cores / 112 threads) and one GPU with 80GB of VRAM. Cache simulation and TPOT post-processing are performed on the same machine on CPU, while the GPU is used for model fine-tuning and for generating routing traces under $B{=}1$ decoding.

**Training hyperparameters and reproducibility.** Unless otherwise stated, all runs use `max_steps=2000`, `seq_len=2048`, `train_bs=1`, `grad_accum=8`, `lr=5e-5`, `warmup_steps=200`, `aux_alpha_train=0.0`, and BF16. For ReMoE, we set `lambda_kl=0.45`, `lambda_reuse=0.2` with `reuse_warmup_steps=400`, and `lambda_smooth=0.05`, `lambda_lag=0.05`, `lambda_ws=0.01` with `loc_warmup_steps=800`. All ablations

keep the setup identical to the full method except for explicitly removed components (e.g., `lambda_kl=0` for w/o Trust, `lambda_reuse=0` for w/o Reuse, and `lambda_smooth=lambda_lag=lambda_ws=0` for w/o Consistency).

**Logging and checkpointing.** We log training metrics every 10 steps (`logging_steps=10`) and run evaluation and check-point saving every 200 steps (`eval_steps=200`, `save_steps=200`). Unless otherwise stated, we disable resuming (`no_resume`) and redirect stdout/stderr to a single log file.

**Training cost.** ReMoE is lightweight to train because only router parameters are updated. Fine-tuning DeepSeek-V2-Lite for 2,000 steps takes approximately 7.9 hours on one 80GB GPU. This cost is paid once at post-training and introduces no additional inference-time parameters or routing logic.

### E.2. Cache Metrics in Our Simulator: uHR and #uMiss

This subsection defines the cache metrics reported by our offline simulator and aligns the notation with the main paper (routed experts $N_r$, Top-$K$ routing with $K$, and per-layer cache capacity $C$). We clarify what is counted at the token level versus the unique (distinct-expert) level, and why "unique" is the right unit for expert-weight I/O.

E.2.1. SETUP: SEGMENTS, REQUESTS, AND CACHE STATE

**Segments and MoE layers.** Let $\mathcal{L}_{\mathrm{MoE}}$ be the set of MoE layers. We simulate decoding in segments (request sessions), indexed by $s \in \{1, \ldots, S\}$, where segment $s$ contains decode steps $t \in \{1, \ldots, T_s\}$. When `--reset_each_batch` is enabled, each batch is treated as one segment and the cache is reset at the start of each segment; otherwise the entire run forms a single segment.

**Top-$K$ expert indices.** At decode step $t$ of segment $s$, for each MoE layer $\ell \in \mathcal{L}_{\mathrm{MoE}}$ and each batch item $b \in \{1, \ldots, B\}$, the router outputs a Top-$K$ expert index list

$$E_{s,t}^{(\ell)}(b) = \left(e_{s,t}^{(\ell)}(b,1), \ldots, e_{s,t}^{(\ell)}(b,K)\right), \qquad e_{s,t}^{(\ell)}(b,k) \in \{1, \ldots, N_r\}.$$

Shared experts (if present) are always executed and are excluded from $E_{s,t}^{(\ell)}(b)$.

**Cache state.** For each layer $\ell$, we maintain a per-layer expert cache with capacity $C$. Let $\mathcal{C}_{s,t}^{(\ell)} \subseteq \{1, \ldots, N_r\}$ denote the resident set (cached experts) before serving step $t$ in segment $s$, with $|\mathcal{C}_{s,t}^{(\ell)}| \leq C$. The update rule depends on the chosen policy (LRU/LFU/FIFO), but the hit/miss definitions below are policy-agnostic.

E.2.2. TOKEN-LEVEL VS. UNIQUE-LEVEL ACCOUNTING

Our simulator reports cache statistics at two granularities.

**Token-level (routing events).** Token-level accounting treats each routed slot as one event, i.e., there are $BK$ events per step for a layer. It measures how many of these routed expert choices are already resident.

**Unique-level (distinct experts per step).** Unique-level accounting deduplicates expert ids within the step and treats each distinct missing expert as one expert-weight fetch. This matches expert offloading: within a decode step, an expert weight tensor (if missing) needs to be loaded at most once even if selected multiple times across batch items. Hence we call it "unique": the unit is the set of distinct requested experts per step, rather than the multiset of routed slots.

E.2.3. PER-STEP HITS AND MISSES

Fix a layer $\ell$ and a step $(s, t)$. Define the flattened multiset (list) of routed expert ids

$$R_{s,t}^{(\ell)} = \left\{e_{s,t}^{(\ell)}(b,k) : b \in \{1, \ldots, B\}, k \in \{1, \ldots, K\}\right\},$$

where $|R_{s,t}^{(\ell)}| = BK$ counting multiplicity. Define the per-step distinct (unique) set

$$U_{s,t}^{(\ell)} = \mathrm{Unique}\left(R_{s,t}^{(\ell)}\right), \qquad |U_{s,t}^{(\ell)}| \leq BK,$$

where $\mathrm{Unique}(\cdot)$ removes duplicates (order-preserving in the implementation).

**Token hits / misses.** We define token-level hits and misses as

$$H_{s,t,\text{tok}}^{(\ell)} = \sum_{e \in R_{s,t}^{(\ell)}} \mathbb{I}\left[e \in \mathcal{C}_{s,t}^{(\ell)}\right], \tag{38}$$

$$T_{s,t,\text{tok}}^{(\ell)} = |R_{s,t}^{(\ell)}| = BK, \tag{39}$$

$$M_{s,t,\text{tok}}^{(\ell)} = T_{s,t,\text{tok}}^{(\ell)} - H_{s,t,\text{tok}}^{(\ell)}. \tag{40}$$

**Unique hits / misses.** We define unique-level hits and misses as

$$H_{s,t,\text{uniq}}^{(\ell)} = \sum_{e \in U_{s,t}^{(\ell)}} \mathbb{I}\left[e \in \mathcal{C}_{s,t}^{(\ell)}\right] = \left|U_{s,t}^{(\ell)} \cap \mathcal{C}_{s,t}^{(\ell)}\right|, \tag{41}$$

$$T_{s,t,\text{uniq}}^{(\ell)} = |U_{s,t}^{(\ell)}|, \tag{42}$$

$$M_{s,t,\text{uniq}}^{(\ell)} = T_{s,t,\text{uniq}}^{(\ell)} - H_{s,t,\text{uniq}}^{(\ell)} = \left|U_{s,t}^{(\ell)} \setminus \mathcal{C}_{s,t}^{(\ell)}\right|. \tag{43}$$

### E.2.4. AGGREGATE METRICS: uHR AND #uMISS

We define the total number of unique misses for layer $\ell$ as

$$\#u\text{Miss}^{(\ell)} = \sum_{s=1}^{S} \sum_{t=1}^{T_s} M_{s,t,\text{uniq}}^{(\ell)} = \sum_{s=1}^{S} \sum_{t=1}^{T_s} \left|U_{s,t}^{(\ell)} \setminus \mathcal{C}_{s,t}^{(\ell)}\right|. \tag{44}$$

We define the unique hit rate (uHR) for layer $\ell$ as

$$u\text{HR}^{(\ell)} = \frac{\sum_{s,t} H_{s,t,\text{uniq}}^{(\ell)}}{\sum_{s,t} T_{s,t,\text{uniq}}^{(\ell)}} = \frac{\sum_{s,t} \left|U_{s,t}^{(\ell)} \cap \mathcal{C}_{s,t}^{(\ell)}\right|}{\sum_{s,t} |U_{s,t}^{(\ell)}|}. \tag{45}$$

Equivalently, $u\text{MR}^{(\ell)} = 1 - u\text{HR}^{(\ell)}$. For completeness, the token hit rate (tHR) is

$$t\text{HR}^{(\ell)} = \frac{\sum_{s,t} H_{s,t,\text{tok}}^{(\ell)}}{\sum_{s,t} T_{s,t,\text{tok}}^{(\ell)}}.$$

### E.2.5. WHY "UNIQUE" MATCHES EXPERT-WEIGHT I/O

Unique misses are designed to match the physical cost of expert offloading. Assume expert weights are fetched in expert-sized blocks and remain resident at least within the current decode step. Then all occurrences of the same expert id in $R_{s,t}^{(\ell)}$ share one underlying weight buffer, and the number of expert-weight fetches needed at step $(s, t)$ is exactly the number of distinct requested experts not currently resident, namely $M_{s,t,\text{uniq}}^{(\ell)}$ in Eq. (43). This motivates using #uMiss (and uHR) as the primary cache metrics when evaluating I/O pressure.

### E.2.6. ACROSS-LAYER AGGREGATION AND STEP-LEVEL REPORTING

**Across-layer aggregation.** We also report overall aggregates by summing counts across MoE layers:

$$\#u\text{Miss}^{(\text{all})} = \sum_{\ell \in \mathcal{L}_{\text{MoE}}} \#u\text{Miss}^{(\ell)}, \qquad u\text{HR}^{(\text{all})} = \frac{\sum_\ell \sum_{s,t} H_{s,t,\text{uniq}}^{(\ell)}}{\sum_\ell \sum_{s,t} T_{s,t,\text{uniq}}^{(\ell)}}.$$

**Per-step (edge) reporting.** For edge decoding, we additionally record per-step aggregates across all MoE layers:

$$\#u\text{Miss}_{s,t}^{(\text{all})} = \sum_{\ell \in \mathcal{L}_{\text{MoE}}} M_{s,t,\text{uniq}}^{(\ell)}, \tag{46}$$

$$H_{s,t,\text{tok}}^{(\text{all})} = \sum_{\ell \in \mathcal{L}_{\text{MoE}}} H_{s,t,\text{tok}}^{(\ell)}, \tag{47}$$

$$T_{s,t,\text{tok}}^{(\text{all})} = \sum_{\ell \in \mathcal{L}_{\text{MoE}}} T_{s,t,\text{tok}}^{(\ell)}. \tag{48}$$

The simulator reports percentiles (P50/P95/P99) of $\#u\text{Miss}_{s,t}^{(\text{all})}$ and token-level hit statistics over decode steps.

**Optional I/O and TPOT estimation.** When `expert_bytes` and `bandwidth_GBps` are provided, the simulator converts per-step unique misses into an estimated I/O time:

$$\text{IO\_ms\_step}(s, t) = \frac{\#u\text{Miss}_{s,t}^{(\text{all})} \cdot \text{expert\_bytes}}{\text{bandwidth}} \times 1000.$$

It further estimates per-token I/O by dividing by $B$ and combines it with a clean measured decode compute baseline to obtain TPOT distributions; these conversions are only for reporting and do not affect the cache hit/miss definitions above.

# F. Additional Experiment Results

### F.1. Full Cache-Simulator Results

*Table 9.* **Expert cache efficiency under request-level resets.** $\Delta$ is ReMoE$-$Baseline; $\Delta\%$ is relative change. #uMiss is reported in millions.

| | (a) Unique hit rate (uHR) ↑ | | | | | | (b) Total unique misses ↓ | | | |
|---|---|---|---|---|---|---|---|---|---|---|
| $C$ | Policy | Base | ReMoE | $\Delta$ | $\Delta\%$ | $C$ | Policy | Base | ReMoE | $\Delta$ | $\Delta\%$ |
| 4 | LRU | 0.2058 | 0.2374 | +0.0316 | +15.36% | 4 | LRU | 1.0150 | 0.9746 | -0.0404 | -3.98% |
| 6 | LRU | 0.3187 | 0.3687 | +0.0500 | +15.69% | 6 | LRU | 0.8707 | 0.8068 | -0.0639 | -7.34% |
| 8 | LRU | 0.3629 | 0.4142 | +0.0513 | +14.14% | 8 | LRU | 0.8141 | 0.7486 | -0.0655 | -8.05% |
| 12 | LRU | 0.4519 | 0.5035 | +0.0516 | +11.42% | 12 | LRU | 0.7005 | 0.6345 | -0.0660 | -9.41% |
| 12 | LFU | 0.4597 | 0.5151 | +0.0554 | +12.05% | 12 | LFU | 0.6904 | 0.6197 | -0.0707 | -10.24% |
| 12 | FIFO | 0.4432 | 0.4930 | +0.0498 | +11.24% | 12 | FIFO | 0.7116 | 0.6479 | -0.0637 | -8.95% |

**Belady's optimal policy.** To separate better routing from better alignment with LRU, we also evaluate Belady's MIN under the same reset protocol. ReMoE yields fewer oracle misses across capacities. For example, at $C{=}4$, oracle misses drop from 843,832 to 802,903 ($\Delta{=}-40{,}929$), and at $C{=}6$ from 684,671 to 635,069 ($\Delta{=}-49{,}602$).

**Step-level TPOT proxy.** We convert step-level unique misses into a simple I/O latency estimate and approximate per-token latency by $\text{TPOT} \approx \text{TPOT}_{\text{compute}} + \text{IO}_{\text{step}}/B$ with $B{=}1$. Using `bandwidth_GBps`=4.0, ReMoE consistently reduces TPOT once the cache is nontrivial. Table 10 reports the resulting step-level TPOT percentiles under LRU.

*Table 10.* **Step-level TPOT percentiles (ms/token, LRU).** $\Delta$ is ReMoE$-$Baseline; $\Delta\%$ is relative reduction. Estimated with `bandwidth_GBps`=4.0 under request-level resets.

| | (a) TPOT$_{50}$ (median) ↓ | | | | | (b) TPOT$_{95}$ ↓ | | | |
|---|---|---|---|---|---|---|---|---|---|
| $C$ | Base | ReMoE | $\Delta$ | $\Delta\%$ | $C$ | Base | ReMoE | $\Delta$ | $\Delta\%$ |
| 4 | 569.0 | 545.3 | -23.7 | -4.2% | 4 | 661.7 | 642.0 | -19.7 | -3.0% |
| 6 | 508.6 | 468.8 | -39.8 | -7.8% | 6 | 649.6 | 617.8 | -31.8 | -4.9% |
| 8 | 476.4 | 436.5 | -39.9 | -8.4% | 8 | 633.5 | 597.7 | -35.8 | -5.7% |
| 12 | 407.9 | 372.1 | -35.8 | -8.8% | 12 | 601.2 | 561.4 | -39.8 | -6.6% |

### F.2. Cache-Aware Inference-Time Rerouting

**Background.** Cache-aware inference-time rerouting methods address the same temporal-locality bottleneck as ReMoE, but operate at a different stage of the deployment pipeline. The most representative work in this line is Mixture of Cache-Conditional Experts (Skliar et al., 2025), which explicitly targets batch-size-one on-device MoE inference, where only a

subset of expert weights fit into DRAM. Their key empirical observation is that MoE routers can tolerate careful deviations in expert selection with only minor predictive-quality loss; building on this, Skliar et al. (2025) introduce a training-free, cache-conditional rerouting rule that, at each decode step, biases router scores toward experts that are already resident in the cache, while still permitting non-resident experts to be selected when their original scores are sufficiently high. On mobile hardware, this is reported to reduce cache miss rates by more than 50% and to deliver up to a $2\times$ end-to-end speedup, with perplexity changes typically in the 0.1%–3% range. Conceptually, the method shifts the quality–locality trade-off entirely to inference time: the underlying model and router are unchanged, and locality is purchased on the fly through a residency-aware re-scoring of the router output.

**Relation to ReMoE.** ReMoE and cache-conditional rerouting are complementary rather than competing. The former reshapes the router through lightweight post-training, so that the produced routing trace is intrinsically more cache-friendly; at inference time, the standard MoE inference graph is preserved and no per-step rerouting machinery is required. The latter modifies the routing decision at serving time, leaving the model weights and router untouched. In principle, the two can be stacked: a router fine-tuned by ReMoE can still be combined with a mild cache-residency bias at inference, and we expect the cache-friendlier base trace produced by ReMoE to make the inference-time bias both safer (smaller deviations from the pretrained policy are sufficient) and more effective.

**Setup.** To compare under the same model and cache budget, we implement a cache-aware rerouting heuristic in the spirit of Skliar et al. (2025) on DeepSeek-V2-Lite, with cache capacity $C{=}4$ and LRU replacement. The heuristic adds a cache-residency bonus, controlled by a strength parameter $\beta$, to the pre-Top-$K$ router scores of experts already in the cache; $\beta{=}0$ recovers the original router (no inference-time intervention), while larger $\beta$ progressively pushes routing decisions toward cached experts. We report unique-expert hit rate (uHR) and LM perplexity (PPL) to expose the quality–locality trade-off.

**Aggressive rerouting hurts quality.** Table 11 shows that strong inference-time rerouting can dramatically lift cache hit rates — uHR rises from 23.74% (ReMoE's learned router, no inference-time intervention) to 63.88% at $\beta{=}4$ — but at the cost of a catastrophic collapse in language modeling quality (PPL $6.35 \rightarrow 3607.92$). Even at the moderate setting $\beta{=}1.0$, PPL already degrades to 10.60. This is consistent with the central caveat of training-free rerouting (Skliar et al., 2025): the tolerance of an MoE router to forced deviations is bounded, and once the bias dominates the original score, routing decisions can no longer recover the model's intended expert–token specialization. ReMoE, by contrast, attains a more stable quality–locality operating point without any runtime rerouting: it sacrifices peak cache hit rate but keeps PPL close to the unperturbed baseline.

*Table 11.* **Aggressive cache-aware inference-time rerouting under** $C{=}4$ **and LRU.** Higher $\beta$ biases routing more strongly toward cached experts. ReMoE reaches a controlled quality–locality operating point without any runtime rerouting, whereas strong inference-time bias rapidly degrades PPL.

| Method | $\beta$ | uHR ↑ | PPL ↓ |
|---|---|---|---|
| ReMoE learned router | 0.0 | 23.74% | 6.35 |
| Baseline + heuristic | 1.0 | 41.66% | 10.60 |
| Baseline + heuristic | 4.0 | 63.88% | 3607.92 |

**Composability with mild rerouting.** We next ask whether ReMoE can be combined with a mild version of cache-conditional rerouting, rather than replaced by it. We fix $\beta{=}0.5$ and apply the same heuristic on top of either the pretrained baseline router or the ReMoE fine-tuned router (Table 12). Under this matched-$\beta$ comparison, ReMoE + heuristic yields higher uHR (34.07% vs. 32.07%), lower PPL (6.51 vs. 6.97), and higher estimated TPS (2.1106 vs. 2.0481) than the baseline + heuristic. In other words, a locality-aware base router gives a mild inference-time bias a better starting point on the trade-off curve: the same amount of rerouting buys more locality with less quality damage when the underlying routing trace is already locality-friendly. This supports our claim that ReMoE is orthogonal to runtime cache-aware routing such as Skliar et al. (2025), rather than a substitute for it.

*Table 12.* **Composability with mild cache-aware rerouting under** $C{=}4$ **and LRU.** At the same heuristic strength $\beta{=}0.5$, ReMoE improves uHR, PPL and estimated TPS over the baseline, indicating that the locality bias internalized by ReMoE during fine-tuning composes constructively with mild inference-time cache-aware routing.

| Method | $\beta$ | uHR ↑ | PPL ↓ | Est. TPS ↑ |
|---|---|---|---|---|
| Baseline + heuristic | 0.5 | 32.07% | 6.97 | 2.0481 |
| ReMoE + heuristic | 0.5 | 34.07% | 6.51 | 2.1106 |

### F.3. Routing Trajectory of DeepSeek-V2-Lite-Chat

**Motivation.** Throughout the main paper, we characterize the short-horizon locality gap on the base DeepSeek-V2-Lite checkpoint. A natural concern is whether this gap is an artifact of the base pretraining stage alone, and whether standard supervised fine-tuning (SFT) or instruction tuning is by itself sufficient to remove it. The question matters in practice because the MoE checkpoints that are actually shipped to end users are almost always chat/instruction-tuned variants rather than raw base models, and SFT is known to perturb both expert utilization and per-token routing distributions. If chat tuning already induced sticky short-horizon reuse on its own, the deployment-time motivation for a dedicated locality-aware adaptation step like ReMoE would be substantially weaker.

**Setup.** To probe this directly, we apply exactly the same teacher-forced routing-trace protocol used in Fig. 2 to the official DeepSeek-V2-Lite-Chat checkpoint, i.e., the SFT/instruction-tuned counterpart of the base model used throughout the main experiments. We feed the same fixed token sequence, record the Top-$K$ expert indices selected by the chat model's router at each decoding step, and visualize the trajectory at the same MoE layer (layer 21) as in Fig. 2, so that the chat and base trajectories are directly comparable. As in the main figure, teacher forcing fixes the input token sequence and the time index, so any difference between trajectories reflects a change in routing policy rather than generation drift.

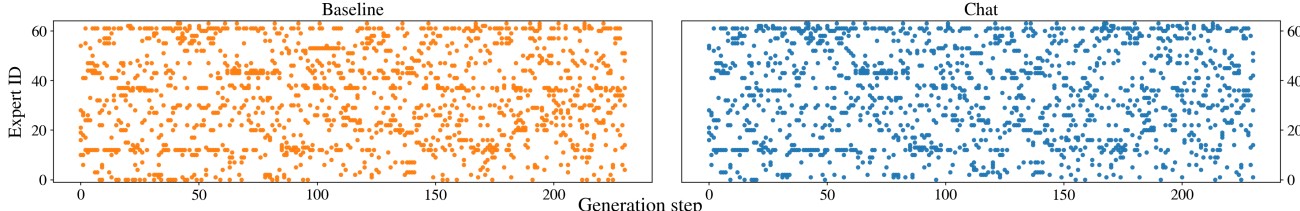

*Figure 4.* **Routing trajectory of DeepSeek-V2-Lite-Chat under teacher forcing (layer 21).** Each point marks one of the Top-$K$ experts chosen at the corresponding decoding step. The chat-tuned model spreads its routing decisions across a wide set of experts and switches selection nearly every step, without the extended horizontal reuse streaks that ReMoE produces on the base model in Fig. 2. Qualitatively, the chat trajectory is closely aligned with the baseline panel of Fig. 2 rather than with the ReMoE panel.

**Observation.** The chat trajectory in Fig. 4 remains token-wise dispersed and exhibits frequent step-to-step expert switches, with no obvious longer reuse streaks. The visual pattern is essentially indistinguishable from the baseline panel of Fig. 2, and clearly different from the locality-stabilized trace produced by ReMoE on the same layer. In short, the short-horizon locality gap identified on the base model carries over to the SFT/chat-tuned model essentially intact.

**Implication.** Two consequences follow for ReMoE's positioning. First, the deployment problem ReMoE targets is real for the model class that is actually deployed: chat/instruction variants inherit the same offloading-unfriendly routing behavior as their base checkpoints, so the cache-locality bottleneck is not bypassed by the standard pretraining→SFT recipe. Second, this is consistent with the CE-only ablation in Table 2 and Table 4, where a router-only continued fine-tuning pass with standard next-token cross-entropy also fails to recover the locality benefit. Together, these two pieces of evidence point in the same direction: generic supervised adaptation, whether at the full-model level (chat/SFT) or restricted to the router (CE-only), does not by itself produce the stable short-horizon expert working set needed for memory-constrained expert offloading. The locality gain reported in this paper is specifically attributable to ReMoE's locality-aware objective, rather than to any router or model adaptation that happens to consume the same data.

## F.4. Generalization Results on Qwen1.5-MoE-A2.7B

*Table 13.* **Generalization on Qwen1.5-MoE-A2.7B:** routing and LM metrics. Rel. $\Delta$ is (ReMoE − Baseline)/Baseline.

| Method | PPL↓ | EOR↑ | Entropy↓ | CV↑ |
|---|---|---|---|---|
| Baseline | 2.7104 | 0.1695 | 0.99996 | 0.0174 |
| ReMoE | 2.3659 | 0.2156 | 0.99861 | 0.1109 |
| Rel. $\Delta$ | −12.7% | +27.2% | −0.14% | +537.4% |

*Table 14.* **Generalization on Qwen1.5-MoE-A2.7B:** downstream benchmarks (lm-eval). Scores are mean $\pm$ stderr reported by `lm_eval`. $\Delta$ is ReMoE−Baseline in percentage points.

| Benchmark (metric) | Baseline | ReMoE | $\Delta$ (pp) |
|---|---|---|---|
| GSM8K (EM, strict) | $16.53 \pm 1.02$ | $18.14 \pm 1.38$ | +1.61 |
| GSM8K (EM, flex) | $60.58 \pm 1.35$ | $61.11 \pm 1.34$ | +0.53 |
| HumanEval (pass@1) | $35.37 \pm 3.74$ | $35.98 \pm 3.76$ | +0.61 |
| MMLU (acc) | $61.10 \pm 0.39$ | $61.20 \pm 0.39$ | +0.10 |

# G. Sensitivity Analysis

This appendix reports a sensitivity study of ReMoE's router-only fine-tuning objective using the same training recipe as in the main paper, while varying (i) the reuse regularizer weight $\lambda_{\text{reuse}}$, (ii) the trust-anchor weight $\lambda_{\text{KL}}$, and (iii) the lag-step set $\mathcal{D}$ used by the temporal-locality objective. All runs are evaluated at the final step (2,000) under the same validation protocol used throughout the paper.

**Default configuration.** Unless otherwise stated, we use $\lambda_{\text{reuse}} = 0.2$, $\lambda_{\text{KL}} = 0.45$, and the default lag set $\mathcal{D}_{\text{main}} = \{1, 2, 4, 8, 16\}$. Other locality-related hyperparameters are kept identical across runs.

## G.1. Sensitivity to $\lambda_{\text{reuse}}$ and $\lambda_{\text{KL}}$

We first vary $\lambda_{\text{reuse}}$ while fixing $\lambda_{\text{KL}} = 0.45$ and $\mathcal{D} = \mathcal{D}_{\text{main}}$, and then vary $\lambda_{\text{KL}}$ while fixing $\lambda_{\text{reuse}} = 0.2$ and $\mathcal{D} = \mathcal{D}_{\text{main}}$. Figure 5 summarizes both sweeps.

**Varying $\lambda_{\text{reuse}}$.** As shown in Figure 5a, increasing $\lambda_{\text{reuse}}$ leads to a consistent increase in the reuse score (`eval_reuse`: $0.283 \rightarrow 0.370$), indicating that $\lambda_{\text{reuse}}$ directly controls expert reuse in this range. Meanwhile, the trust-anchor deviation (`eval_trust_kl`) increases with larger $\lambda_{\text{reuse}}$ ($0.0098 \rightarrow 0.0667$), reflecting a larger distributional drift from the frozen reference router. Across the sweep, language-model validation metrics remain stable (PPL $\approx$ 3.22–3.24; Acc@1 $\approx$ 71.7–71.8), suggesting that the reuse gain is achieved without degrading capability in these runs.

**Varying $\lambda_{\text{KL}}$.** Figure 5b shows that $\lambda_{\text{KL}}$ strongly controls the router's distributional drift: removing the anchor ($\lambda_{\text{KL}} = 0$) yields a large trust deviation (`eval_trust_kl`=0.308) and slightly worse PPL (3.264), while increasing $\lambda_{\text{KL}}$ further reduces the match to the reference router (`eval_trust_kl` decreases to 0.016 at $\lambda_{\text{KL}} = 0.7$). Consistent with the anchor constraining optimization freedom, reuse decreases as $\lambda_{\text{KL}}$ grows (`eval_reuse`: 0.386 at $\lambda_{\text{KL}} = 0$ to 0.321 at $\lambda_{\text{KL}} = 0.7$). Overall, $\lambda_{\text{KL}} = 0.45$ yields strong reuse gains with moderate drift in this setting.

## G.2. Sensitivity to the lag-step set $\mathcal{D}$ (short-lag variant)

We compare the default lag set $\mathcal{D}_{\text{main}} = \{1, 2, 4, 8, 16\}$ against a shorter set $\mathcal{D}_{\text{short}} = \{1, 2, 4, 8\}$, keeping $\lambda_{\text{reuse}} = 0.2$ and $\lambda_{\text{KL}} = 0.45$ fixed. Since the comparison involves only two configurations, we summarize the results in Table 15. All metrics remain extremely close between the two settings, indicating that replacing $\mathcal{D}_{\text{main}}$ by $\mathcal{D}_{\text{short}}$ does not materially change the outcome in this experiment, and the dominant effect is already captured by short-horizon constraints in $\{1, 2, 4, 8\}$.

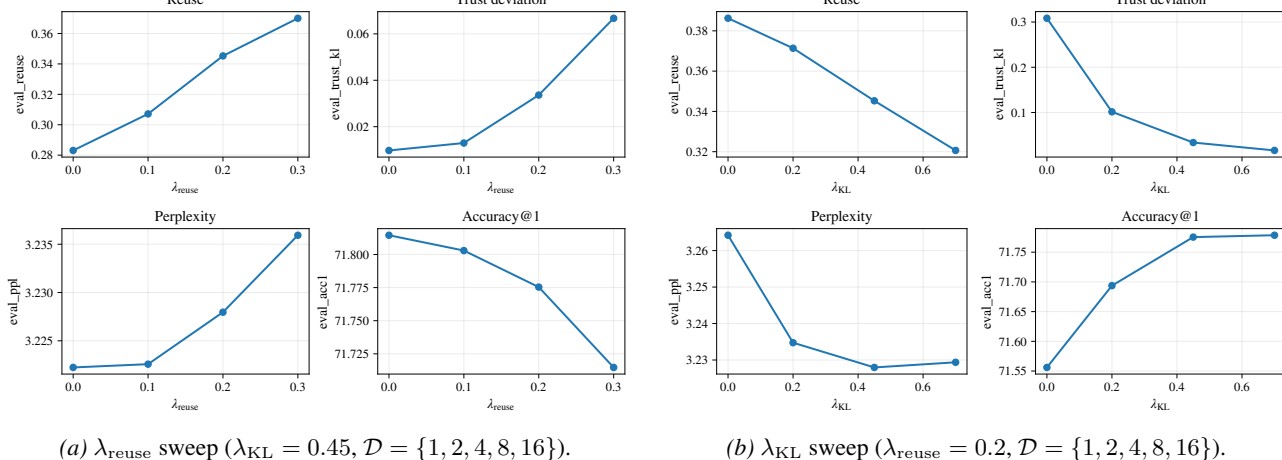

*(a)* $\lambda_{\mathrm{reuse}}$ sweep ($\lambda_{\mathrm{KL}} = 0.45$, $\mathcal{D} = \{1, 2, 4, 8, 16\}$).    *(b)* $\lambda_{\mathrm{KL}}$ sweep ($\lambda_{\mathrm{reuse}} = 0.2$, $\mathcal{D} = \{1, 2, 4, 8, 16\}$).

*Figure 5.* **Sensitivity to** $\lambda_{\mathrm{reuse}}$ **and** $\lambda_{\mathrm{KL}}$. Increasing $\lambda_{\mathrm{reuse}}$ improves reuse (EOR; reported as `eval_reuse`) at the cost of larger trust deviation (`eval_trust_kl`), while increasing $\lambda_{\mathrm{KL}}$ reduces drift but also constrains reuse.

| Metric | Default $\mathcal{D}_{\mathrm{main}}$ | Short $\mathcal{D}_{\mathrm{short}}$ | Ratio (Short / Default) |
|---|---|---|---|
| PPL (`eval_ppl`) | 3.2280 | 3.2298 | 1.0006 |
| Acc@1 (`eval_acc1`) | 71.775 | 71.781 | 1.0001 |
| Reuse (`eval_reuse`) | 0.3453 | 0.3440 | 0.9963 |
| Trust dev. (`eval_trust_kl`) | 0.03363 | 0.03320 | 0.9872 |
| CV (`eval_cv`) | 0.16085 | 0.15943 | 0.9912 |

*Table 15.* **Sensitivity to the lag-step set** $\mathcal{D}$. Comparison between the default lag set $\mathcal{D}_{\mathrm{main}} = \{1, 2, 4, 8, 16\}$ and the short-lag set $\mathcal{D}_{\mathrm{short}} = \{1, 2, 4, 8\}$ with $\lambda_{\mathrm{reuse}} = 0.2$ and $\lambda_{\mathrm{KL}} = 0.45$. Ratios close to 1.0 indicate minimal change.

