# OpenReview forum: "ReMoE: Boosting Expert Reuse through Router Fine-Tuning in Memory-Constrained MoE LLM Inference"
_ICML.cc/2026/Conference — ICML 2026 regular_

### Official Review · Reviewer_gQ3Q · 2026-02-24

**Soundness:** 3
**Presentation:** 2
**Significance:** 3
**Originality:** 3
**Overall Recommendation:** 3
**Confidence:** 3

**Summary:**

This paper proposes a fine-tuning approach for Mixture-of-Experts (MoE) models with the aim of improving expert reuse in autoregressive token generation. In this fine-tuning approach, all of the model parameters except the routers are frozen. The model is fed input prompts, and the router is expected to route the hidden states to the experts that have already been used for the calculation of previous tokens.  By doing this, during inference, the fine-tuned model can reuse already-available experts cached on the GPU in memory-constrained settings. This reduces generation latency and increases the requests handled per unit of time by the serving system.

**Compliance With Llm Reviewing Policy:**

Affirmed.

**Final Justification:**

My concerns are not fully resolved. As mentioned in rebuttal ack, I will keep my current rating.

**Key Questions For Authors:**

- How does your post-training affect the baseline? Would the baseline achieve improved performance if its routers were fine-tuned?
- How do you justify the improved accuracy of the generated outputs on certain tasks? Based on my understanding, you are modifying the original model.
- Although inference latency and throughput are provided, only the number of fine-tuning steps is reported for the training phase. How much time does each step take?
- Since you are improving the temporal locality of experts, which is dependent on the input prompt, does your approach require further fine-tuning to adapt to other input domains?

**Limitations:**

Yes.

**Strengths And Weaknesses:**

### Strengths:

- The proposed approach is well-defined and well-presented.
- The authors have evaluated the quality of the generated output of their model using multiple benchmarks. The ablation study assesses the impact of each part of the approach on the results.
- The paper aims to address the challenges of MoE inference in memory-constrained settings, which is useful for on-device deployment of edge devices.
- Although inference optimizations such as expert caching have been extensively explored in prior research, fine-tuning the router with this goal seems to be novel.

### Weaknesses:

- As mentioned by the authors, this fine-tuning leads to expert imbalance. Load balancing is a key principle in conventional MoE training; however, the approach proposed in this paper is at odds with this principle.
- There is no analysis of inference expert imbalance in the manuscript.
- The proposed fine-tuning also reduces the model’s expressiveness by limiting the number of possible expert combinations. Moreover, the proposed temporal inductive bias makes the model vulnerable to semantic shifts, such as a sudden transition from conversational English to a complex Python script.

---

> ### Author Rebuttal · Authors · 2026-03-31
>
> Thank you for your professional review comments and suggestions. We have added results and will include them in revision.
>
> **Q1. How does the post-training affect the baseline? Would the baseline achieve improved performance if its routers were fine-tuned?**
>
> **A1.** ReMoE reshapes MoE routing to be more cache-friendly without modifying expert parameters or the inference runtime.
> We freeze all non-router weights and fine-tune only the gate parameters $\theta_{\text{gate}}$.
>
> We added a CE-only router fine-tuning baseline: only optimized with standard LM cross-entropy, without ReMoE’s locality objective. CE-only does not reproduce the locality gain, and the same pattern appears in downstream evaluation. Thus, the gain cannot be explained by router adaptation alone; the locality-aware objective is necessary.
>
> | Metric         | CE-only | Baseline | ReMoE  |
> | -------------- | ------- | -------- | ------ |
> | EOR            | 0.2293  | 0.2730   | 0.3453 |
> | GSM8K (strict) | 36.92   | 38.89    | 38.13  |
> | GSM8K (flex)   | 37.23   | 39.04    | 38.36  |
> | HumanEval      | 28.05   | 26.83    | 29.27  |
> | MMLU           | 57.44   | 57.72    | 57.81  |
>
> Using vLLM (PR #37190; per-layer GPU LRU cache with CPU offloading), max-num-seqs=1, moe-expert-cache-size=6, and ShareGPT prompts (concurrency = 1), ReMoE improves inference efficiency:
>
> | Method    | Output token throughput (tok/s) ↑ | Mean TTFT (ms) ↓ | Mean TPOT (ms) ↓ | Avg. per-layer unique-expert hit rate ↑ |
> | --------- | --------------------------------- | ---------------- | ---------------- | --------------------------------------- |
> | Baseline  | 3.58                              | 769.23           | 254.31           | 39.4%                                   |
> | CE-only   | 2.95                              | 780.12           | 286.82           | 21.1%                                   |
> | **ReMoE** | **3.88**                          | **758.27**       | **242.99**       | **43.3%**                               |
>
> **Q2. There is no analysis of inference expert imbalance in the manuscript.**
>
> **A2.** We agree this should be clarified. The “imbalance” is controlled and is beneficial. In our target setting, a moderate increase in inference-time expert imbalance is expected and beneficial, because it improves short-horizon reuse and cache efficiency.
>
> ReMoE increases EOR from 27.3% to 34.5%, indicating stronger adjacent-step expert reuse. At the same time, routing entropy remains near 1 (0.9998 → 0.9971), meaning routing is still globally spread across experts rather than collapsing to a few. The load-balance CV rises from 0.0409 to 0.1608, indicating some increase in usage unevenness, but still in a controlled range. Moreover, unique_experts_per_seq changes only from 64.000 to 63.997, showing that the model still visits essentially the same number of distinct experts over a sequence.
>
> **Q3. How is the improved accuracy of the generated outputs on certain tasks justified?**
>
> **A3.** We interpret the gains on some tasks as a byproduct of better routing, rather than the primary objective of the method. ReMoE updates only the router, with all expert weights frozen, so any output change must come from different expert selection rather than altered expert knowledge. In a fine-grained MoE, experts are heterogeneous, and routing decisions near Top-K boundaries may be noisy. By reducing routing jitter while remaining anchored to the pretrained router, ReMoE can occasionally improve expert-task matching, i.e., activate experts that are already better suited to certain token patterns or subskills.
>
> **Q4. Does enforcing temporal expert reuse reduce routing expressiveness or hurt robustness under abrupt semantic shifts?**
>
> **A4.** No. ReMoE adds a soft locality bias, not a hard reuse constraint: expert switching is still allowed when semantics favor it. Trust-KL anchors the updated router to the pretrained routing distribution, so locality is encouraged only insofar as it remains compatible with the original routing behavior. Removing Trust-KL increases EOR (0.3453 → 0.3877), but also makes routing more concentrated (entropy: 0.9971 → 0.9950; CV: 0.1608 → 0.2110) and slightly worsens LM quality (PPL: 3.2280 → 3.2629; Acc@1: 71.78 → 71.58).
>
> **Q5. What is the actual training overhead of ReMoE in wall-clock time?**
>
> **A5.** ReMoE takes about 7.9 hours for 2,000 steps on one NVIDIA A100-SXM4-80GB GPU.
>
> **Q6. Does the approach require further fine-tuning to adapt to other domains?**
>
> **A6.** It depends on the target domain. ReMoE is conducted on a multi-domain corpus, so the learned routing-locality bias is intended to transfer across a range of common domains. For domains that are clearly out of distribution with respect to both pretraining and ReMoE fine-tuning data, further fine-tuning would likely be needed, since the desired routing behavior still depends on domain-specific token and expert-usage patterns.

---

> > ### Author Rebuttal · Reviewer_gQ3Q · 2026-04-03
> >
> > I appreciate your rebuttal and further clarification.
> >
> > The rebuttal confirms a trade-off between efficiency and expressiveness; the increase in load-balance CV suggests a routing distortion that favors cache hits over potentially more optimal expert selections. This imbalance will be further concerning for smaller MoE models where expert capacity is limited, as the unevenness could lead to more pronounced performance degradation than observed in the model tested.
> >
> > Furthermore, my concerns regarding robustness during abrupt in-context semantic shifts and the requirement for domain-specific fine-tuning remain unresolved. The authors acknowledge that out-of-distribution domains would likely necessitate further tuning, which restricts the practical utility of the method.

---

> > > ### Author Response · Authors · 2026-04-07
> > >
> > > Thank you for your continued engagement and constructive feedback.
> > >
> > > **Q1. Does ReMoE hurt robustness under abrupt semantic shifts?**
> > >
> > > **A1.** ReMoE mainly reduces routing jitter around the pretrained router’s decision boundary, instead of enforcing reuse regardless of context. This distinction matters because router outputs already encode semantics to a substantial extent: they reflect how the current hidden state is mapped to expert preferences. In ReMoE, locality is introduced only as a soft short-horizon bias, while the Trust-KL anchor is computed against the frozen pretrained router evaluated on the current hidden state. As a result, when the context changes, the reference routing distribution changes as well, and the updated router is still encouraged to follow the pretrained semantic routing for the new context rather than to persist with the previous experts. This interpretation is also consistent with our ablation: removing Trust-KL yields higher reuse, but also more concentrated routing and slightly worse LM quality.
> > >
> > > To directly probe this mode, we additionally ran a controlled teacher-forced semantic-shift diagnostic using explicitly constructed cross-domain segment boundaries (e.g., chat→Python). Two observations are consistent. First, in the boundary-centered expert-overlap plot, ReMoE tracks the baseline closely around the constructed boundary and does not exhibit abnormally persistent overlap there, suggesting that it is not simply “stuck” on the previous experts when the segment changes; at the same time, it maintains higher overlap across most positions on both sides of the boundary, indicating that its main effect is to improve within-segment routing stability rather than to suppress necessary cross-segment switching. Second, in the cross-model boundary-centered JS-similarity plot, the routing distributions of ReMoE and the baseline remain broadly similar across most relative positions, with only a localized dip near the semantic-shift boundary. This indicates that ReMoE does not induce a uniform global drift in routing semantics; instead, its deviation from the pretrained router is structured and concentrated near the shift region, where a routing adjustment is most expected. Taken together, these results suggest that ReMoE preserves robustness under abrupt semantic shifts while mainly improving short-horizon reuse away from the boundary. Figures can be accessed at: https://anonymous.4open.science/r/icml-26-rebuttal/boundary_ir.png and https://anonymous.4open.science/r/icml-26-rebuttal/boundary_similarity.png.
> > >
> > > **Q2. Does ReMoE require domain-specific fine-tuning?**
> > >
> > > **A2.** ReMoE is not designed as a domain-specific adaptation method. Its core contribution lies in the router-level objective design: it encourages short-horizon expert reuse while anchoring the updated router to the pretrained routing distribution through Trust-KL. In this sense, the method operates on routing behavior, not on domain-specific content knowledge, and is better understood as a deployment-aware routing objective rather than a recipe that must be re-tuned for each input domain.
> > >
> > > Since ReMoE freezes all expert weights and constrains the updated router through Trust-KL, the more natural failure mode under distribution shift is that the locality bias becomes less aligned with the routing pattern of that domain, so the cache-efficiency gain may attenuate toward the baseline level. This is different from evidence that the model's knowledge itself has been altered in a way that systematically harms quality: expert weights remain frozen, and the Trust-KL anchor limits arbitrary drift away from the pretrained routing behavior. A more precise limitation statement is therefore that ReMoE's efficiency gain may not fully transfer to strongly OOD domains, rather than that domain-specific re-tuning is inherently required to maintain model quality.
> > >
> > > More importantly, when domain-specific adaptation is genuinely required, ReMoE does not introduce additional fine-tuning cost relative to standard SFT. The only difference between ReMoE and a conventional fine-tuning pass is the router-level locality objective; the training procedure, data format, and compute budget are otherwise identical. In practice, if a new target domain warrants fine-tuning at all — as it would for any SFT, LoRA, or adapter-based method — that single fine-tuning pass simultaneously achieves domain adaptation and routing-locality optimization. ReMoE therefore does not impose an extra tuning stage on top of what would already be required; the locality benefit comes for free within the same fine-tuning budget that domain adaptation already demands.

---

### Official Review · Reviewer_Egjq · 2026-03-06

**Soundness:** 3
**Presentation:** 3
**Significance:** 2
**Originality:** 3
**Overall Recommendation:** 4
**Confidence:** 2

**Summary:**

This paper proposes ReMoE, a lightweight router fine-tuning framework designed to boost token-wise expert reuse. ReMoE aims to address the issue of frequent expert switching in MoE models during edge deployment. By designing specific training objectives, the framework improves the temporal consistency of MoE routing while anchoring the updated router to its original semantics. The authors conduct experiments demonstrating that ReMoE improves inherent cache efficiency, thereby increasing throughput while preserving performance on downstream tasks.

---
Reviewer's Note:

As a point of clarification, I am not familiar with the efficiency metrics and background knowledge regarding MoE edge deployment. My assessment is primarily focused on the MoE routing design and the overall clarity and presentation of the paper. I would like the ACs to take this into account when weighing my review.

**Compliance With Llm Reviewing Policy:**

Affirmed.

**Final Justification:**

Most of my concerns have been addressed, I will maintain my current positive rating.

**Key Questions For Authors:**

- Q1: What is the performance of the DeepSeek-V2-Lite fine-tuned on OpenHermes 2.5 in Table 4 ?

Since ReMoE is trained on OpenHermes 2.5, including the SFT results for DeepSeek-V2-Lite would be more convincing to show that improvements are not merely due to the data. Alternatively, the experiments in Table 4 could be replicated using domain-agnostic pre-training data. Additionally, the authors could include a further ablation study in Tables 4 and 5, where the router is only fine-tuned with CE.

- Q2: Could you reproduce Figure 2 using DeepSeek-V2-Lite-Chat and include it in future revision ?

SFT MoE models exhibit routing patterns that are inconsistent with those of base MoE models. Including a discussion on chat-tuned models would make the conclusions more robust and comprehensive.

**Limitations:**

yes

**Strengths And Weaknesses:**

---
> Strengths

- EMoE is well-motivated; the authors clearly pinpoint the bottlenecks in existing MoE edge deployment and naturally introduce ReMoE as an effective solution to these challenges.

- The authors conduct comprehensive experiments to demonstrate the effectiveness of EMoE.

- This paper is well-written and easy to follow.

---
> Weaknesses

- My primary concern is the necessity of deploying MoE in memory-constrained edge-devices. I appreciate the authors' clarifications on this point; however, it remains unclear whether MoE models can truly remain competitive with SOTA dense models at such a limited scale.

- Some experimental setups regarding the downstream performance can be refined to enhance the overall soundness of this paper. (See Questions)

- ReMoE introduces complex hyperparameter setups. Although the authors demonstrate the insensitivity of these hyperparameters by transferring them across base models without tuning, concerns remain as to whether this set of parameters remains effective across models of different scales and varying activation configurations. At the very least, a sensitivity analysis on the current architecture should be included.

---

> ### Author Rebuttal · Authors · 2026-03-31
>
> Thank you for your professional review comments and suggestions. We have added results and will include them in revision.
>
> **Q1. Is deploying MoE on memory-constrained edge devices truly necessary?**
>
> **A1.** Yes, MoE has become a realistic and increasingly competitive candidate for memory-constrained deployment. First, public results suggest that MoE can remain competitive under constrained runtime-compute and memory budgets. Qwen1.5-MoE-A2.7B is reported to match strong 7B dense baselines while 1.74× faster. This does not prove MoE always wins on edge, but it shows MoE is already a serious candidate. Second, memory-constrained offloading is already a concrete deployment scenario. Samsung UFS 4.0 provides up to 1TB capacity and 4200 MB/s sequential read, and recent systems such as *LLM in a Flash* and *PowerInfer-2* study serving models larger than available DRAM by loading weights from storage on demand. Third, our paper targets the main bottleneck in this setting. Under memory-constrained, batch-size-1 inference, token-wise expert switching causes irregular expert-weight I/O. ReMoE addresses this through router-only fine-tuning with no extra inference-time overhead.
>
> **Q2. What is the performance of the DeepSeek-V2-Lite fine-tuned on OpenHermes 2.5 in Table 4?**
>
> **A2.** We added a CE-only router fine-tuning baseline: only optimized with standard LM cross-entropy, without ReMoE’s locality objective. CE-only does not reproduce the locality gain, and the same pattern appears in downstream evaluation. Thus, the gain cannot be explained by router adaptation alone; the locality-aware objective is necessary.
>
> | Metric         | CE-only | Baseline | ReMoE  |
> | -------------- | ------- | -------- | ------ |
> | EOR            | 0.2293  | 0.2730   | 0.3453 |
> | GSM8K (strict) | 36.92   | 38.89    | 38.13  |
> | GSM8K (flex)   | 37.23   | 39.04    | 38.36  |
> | HumanEval      | 28.05   | 26.83    | 29.27  |
> | MMLU           | 57.44   | 57.72    | 57.81  |
>
> Using vLLM (PR #37190; per-layer GPU LRU cache with CPU offloading), max-num-seqs=1, moe-expert-cache-size=6, and ShareGPT prompts (concurrency = 1), ReMoE improves inference efficiency:
>
> | Method    | Output token throughput (tok/s) ↑ | Mean TTFT (ms) ↓ | Mean TPOT (ms) ↓ | Avg. per-layer unique-expert hit rate ↑ |
> | --------- | --------------------------------- | ---------------- | ---------------- | --------------------------------------- |
> | Baseline  | 3.58                              | 769.23           | 254.31           | 39.4%                                   |
> | CE-only   | 2.95                              | 780.12           | 286.82           | 21.1%                                   |
> | **ReMoE** | **3.88**                          | **758.27**       | **242.99**       | **43.3%**                               |
>
> **Q3. Could the authors reproduce Figure 2 using DeepSeek-V2-Lite-Chat and include it in future revision?**
>
> **A3.** Yes. We also inspected DeepSeek-V2-Lite-Chat under the same teacher-forced setting. Qualitatively, its routing trajectory is similar to the base model—still dispersed and frequently switching, without obvious longer reuse streaks—suggesting that chat/SFT tuning does not by itself remove the same short-horizon locality issue. Figures can be accessed at: https://anonymous.4open.science/r/icml-26-rebuttal/trajectory.tf.layer21.png
>
> **Q4. How sensitive is ReMoE to its hyperparameter choices, and can the proposed hyperparameter setting remain effective across models with different scales and activation configurations?**
>
> **A4.** ReMoE is not highly sensitive within the tested range, and the same recipe remains effective across the two tested MoE families without retuning.
>
> We provide two pieces of evidence. First, Appendix G shows smooth trade-offs: increasing λreuse consistently improves reuse, increasing λKL predictably reduces routing drift while constraining reuse, and replacing the lag set causes only negligible change. Second, the same gate-only recipe transfers without retuning from DeepSeek-V2-Lite to Qwen1.5-MoE-A2.7B, improving EOR from 0.1695 to 0.2156 (+27.2%) while keeping downstream capability comparable(Sec. 5.5 and Tables 8–9 in Appendix F.2 for full results). We do not claim one setting is universally optimal for all MoE scales and activation patterns, but the current results support stable sensitivity on the studied architecture and zero-shot transferability across the tested MoE families.

---

> > ### Author Rebuttal · Reviewer_Egjq · 2026-04-03
> >
> > We appreciate the authors' rebuttal; most of my concerns have been addressed. My remaining comments are as follows:
> >
> > 1) It is my understanding that, foundation model providers may not prioritize small-scale MoE development for memory-constrained settings. As a result, SOTA dense models likely remain more capable in these specific scenarios. A similar concern was also mentioned by Reviewer bHo9
> >
> > 2) While I was seeking a single setting that is universally optimal across all MoEs, I consider this point partially resolved, given that the cost of hyperparameter search is relatively manageable.
> >
> > Thanks for the authors' response, I will maintain my current positive rating.

---

> > > ### Author Response · Authors · 2026-04-07
> > >
> > > Thank you for your continued engagement and constructive feedback.
> > >
> > > **Q1. SOTA dense models likely remain more capable in these specific scenarios.**
> > >
> > > **A1.** We thank the reviewer for this point. Our motivation is that edge/resource-constrained MoE is no longer a purely hypothetical setting, and thus its deployment bottlenecks are timely to study.
> > >
> > > 1. [OPPO](https://www.oppo.com/en/newsroom/press/oppo-leads-ai-innovation-with-on-device-moe/) announced its claimed world’s first on-device MoE implementation, stating that it collaborated with leading chipset providers and that internal lab tests showed about 40% faster AI task processing together with improved energy efficiency. One of their chip providers [MediaTek](https://www.mediatek.com/products/smartphones/mediatek-dimensity-9400) states that its Dimensity 9400 / 9400+ platforms support hybrid MoE and latest MoE model support.
> > > 2. [NVIDIA](https://developer.nvidia.com/blog/build-next-gen-physical-ai-with-edge%E2%80%91first-llms-for-autonomous-vehicles-and-robotics/)’s latest TensorRT Edge-LLM release fully enables MoE support at the edge and targets resource-constrained platforms such as Jetson and DRIVE Thor, positioning MoE as a practical way to deliver higher-capability reasoning under strict latency and power budgets.
> > > 3. [Liquid AI](https://www.liquid.ai/blog/lfm2-8b-a1b-an-efficient-on-device-mixture-of-experts) released LFM2-8B-A1B, positioned for phones, tablets, and laptops; the official release and technical report show it matches the quality of 3–4B dense models while achieving 2.8–3.7× higher decode throughput than dense 4B baselines (e.g., Qwen3-4B) on real consumer CPU targets (Snapdragon SoC on Galaxy S24 Ultra and AMD Ryzen HX370).
> > > 4. [AllenAI](https://allenai.org/blog/olmoe-app) released OLMoE as a fully open-source on-device app; its official materials state that it runs completely on-device, requires no cloud connectivity, and reaches 41 tokens/s on an iPhone 16 Pro.
> > > 5. [Google](https://ai.google.dev/gemma/docs/core/model_card_4) states that Gemma 4’s 26B A4B MoE runs almost as fast as a 4B-parameter dense model while delivering quality within ~2% of the much larger 31B dense model across benchmarks, making it a strong fast-inference alternative to the dense 31B model.
> > >
> > > In summary, our motivation is to address a concrete and increasingly relevant deployment problem: when fine-grained MoE is deployed under limited fast memory, expert switching turns offloading into a first-order latency bottleneck. ReMoE targets this bottleneck directly by making routing more cache-friendly, while remaining complementary to system-level optimizations.
> > >
> > > **Q2. How practically manageable is ReMoE’s hyperparameter tuning across different MoE settings?**
> > >
> > > **A2.** We agree that our current results do not establish a universally optimal hyperparameter setting across all MoE architectures, scales, and activation patterns. Our point is that ReMoE has a small, structured, and manageable tuning burden.
> > >
> > > Our sensitivity results suggest that tuning is effectively low-dimensional. `λreuse` mainly controls locality / expert reuse, while `λKL` mainly controls semantic drift relative to the frozen router. In contrast, the other locality terms (`λsmooth`, `λlag`, `λws`) and the lag-step set have much smaller impact once set to reasonable defaults. Appendix G already shows smooth and interpretable trends: increasing `λreuse` consistently improves reuse, increasing `λKL` predictably constrains drift while reducing reuse, and changing the lag set has negligible effect.
> > >
> > > This yields a simple practical protocol rather than a large unconstrained search: start from the transferred default recipe, keep the secondary locality terms and warmup schedule fixed, sweep only the two primary coefficients (`λreuse`, `λKL`), and select the strongest-locality checkpoint among runs with stable validation quality rather than maximizing EOR in isolation. In this sense, tuning is effectively a small 1D trade-off sweep between locality gain and routing drift.
> > >
> > > ReMoE shows stable, interpretable sensitivity on the studied architecture, transfers across the two tested MoE families without retuning, and requires only a practically manageable low-dimensional search when further tuning is needed.

---

### Official Review · Reviewer_7YV6 · 2026-03-10

**Soundness:** 3
**Presentation:** 4
**Significance:** 3
**Originality:** 2
**Overall Recommendation:** 5
**Confidence:** 5

**Summary:**

This paper investigates the problem of autoregressive inference for MoE LLMs in memory-constrained edge scenarios with a batch size of one. The authors point out that in such environments, expert weights cannot reside entirely in fast memory (RAM), requiring a portion of experts to be loaded from slow storage. Standard token-level routing causes high-frequency switching of activated experts between adjacent tokens, leading to frequent expert I/O, which becomes the primary bottleneck for inference latency. To address this, the paper proposes ReMoE: a post-training method that fine-tunes only the router/gate. By introducing temporal locality regularization, the method encourages adjacent tokens to reuse the same or similar experts. It also employs a Trust-KL constraint to ensure the new router’s output does not deviate too far from the original, thereby preserving the model's inherent capabilities. The authors conducted router-only fine-tuning on DeepSeek-V2-Lite and Qwen1.5-MoE-A2.7B. Through routing trace simulations, cache hit/miss analysis, and language task evaluations, they demonstrate that ReMoE increases the expert overlap ratio and reduces unique cache misses. This results in gains in proxy latency metrics while maintaining baseline performance levels.

**Compliance With Llm Reviewing Policy:**

Affirmed.

**Final Justification:**

As mentioned in rebuttal ack.

**Key Questions For Authors:**

1. Could the authors include a "router CE-only continued fine-tuning" baseline to isolate the net contribution of ReMoE's locality regularization?
2. Is it possible to provide a more direct comparison with Oracle-MoE or Mixture of Cache-Conditional Experts? At a minimum, it would be highly valuable to compare locality, quality, and latency under the same model architecture or a comparable cache budget.

**Limitations:**

I believe the strengths of this paper are that it addresses a real-world problem, proposes a lightweight method, is clearly written, and provides a relatively honest disclosure of its assumptions. Additionally, it demonstrates credible gains at the levels of routing traces and cache simulation. For the post-training adaptation of existing open-source MoE models, ReMoE represents a practically attractive approach. However, judged against formal acceptance criteria, the current chain of evidence remains incomplete. The manuscript lacks robust comparisons with the latest and most relevant literature on this topic. Furthermore, the attribution of the root cause and the theoretical explanations involve a certain degree of extrapolation. For these reasons, I am currently leaning towards a Weak Reject. If the authors can provide the missing key baselines during the rebuttal, I would be willing to raise my score to a 5.

**Strengths And Weaknesses:**

Strengths
1. The topic selection of this paper is both practical and important. For batch-size-one edge/mobile/memory-constrained MoE inference, the temporal inconsistency of expert activation and expert swapping are indeed issues that the research community is actively addressing; this aligns with the direction of recently published relevant literature.

2. The method design is relatively straightforward. ReMoE freezes the embeddings, attention blocks, and expert FFNs, updating only the gate parameters, and optimizes the router together with the language modeling loss, Trust-KL, and a set of locality regularizers. Such a design has clear engineering appeal: it requires low training costs, does not alter the inference graph during deployment, and does not introduce additional runtime logic. In terms of "retrofitting existing open-source MoE," the approach proposed in this paper is practical.

3. The results of the paper, based on trace-level evidence, are clear. The authors report that the EOR improved from 27.3% to 34.5%, representing a relative increase of 26.4%; with C=6 and LRU, uHR increased from 0.3187 to 0.3687, and #uMiss decreased from 0.8707M to 0.8068M; corresponding proxy latency/throughput analysis showed that TPOT50 decreased by 7.8%, while proxy throughput increased by 8.5%.

4. Additionally, the paper is relatively honest about revealing theoretical limitations. The authors clearly state that the relationship between EOR and the upper bound on fetch depends on conditions such as 𝐶≥𝐾, standard serve-and-admit behavior, per-layer cache isolation, and no inter-step interference, and discuss in the appendix when these assumptions fail. This approach is more rigorous than many papers that provide only intuition without explaining the boundaries of their failures.

Weakness
1. The paper's academic positioning relative to the most relevant prior work remains insufficient. It is particularly worth pointing out two works officially published in 2025. Oracle-MoE[1] similarly attributes the core bottleneck of memory-constrained MoE inference to the temporal inconsistency of inter-token expert activations, and improves expert activation consistency and inference speed through a locality-preserving routing architecture. Mixture of Cache-Conditional Experts[2] targets batch-size-one mobile/on-device inference where only a subset of experts can reside in DRAM. It proposes a training-free cache-aware routing method, reporting an over 50% reduction in cache misses and a 2× speedup on mobile hardware. Although the intervention levels of these two works differ from this paper—one leaning toward architecture redesign, the other toward inference-time rerouting, while ReMoE focuses on training-time router adaptation—all three clearly belong to the same problem family and share highly overlapping objectives. The related work section of the current manuscript explicitly discusses offloading/caching system works such as MoE-Infinity, HOBBIT, and FineMoE, and positions ReMoE as a method that "reshapes the trace." Given this context, I believe the authors should, at a minimum, explicitly discuss Oracle-MoE and Cache-Conditional Experts, and ideally provide more direct empirical comparisons.
2. The baseline comparisons remain incomplete. A crucial missing control is a baseline that performs only router continued fine-tuning without incorporating the locality regularizers. Under the current experimental setup, the baseline is the original pretrained router rather than a "CE-only router adaptation." Consequently, the existing results cannot fully disentangle whether the performance gains stem from the locality objective introduced by ReMoE, or merely from generic router adaptation to the OpenHermes-2.5 training distribution itself. This gap must be isolated and rigorously verified, particularly when the paper reports certain improvements in perplexity (PPL).
3. There remains a noticeable surrogate gap between the theoretical analysis and the proposed methodology. While the paper's proposition regarding the Expert Overlap Ratio (EOR) and the number of fetches is meaningful under its specific assumptions, what ReMoE actually optimizes is a differentiable surrogate based on probability mass, rather than the deterministic top-k overlap itself. Furthermore, the Trust-KL objective merely constrains the output distribution of the trainable router to closely match that of the frozen router given the current hidden state; it does not provide a strong guarantee of end-to-end behavior preservation. Therefore, the theoretical portion of this manuscript serves more as a supporting analysis and rationalization of the approach, rather than a rigorous proof of its core claims.

[1] Zhou, J., Dong, F., Huang, R., Cao, H., Chen, M., Yang, Y., Chen, A., Dong, M., Wang, Y., Li, D., Clifton, D. A., Lv, Q., Zhu, R., Zhang, C., Yang, F., Lu, T., Gu, N., & Shang, L. (2025). Oracle-MoE: Locality-preserving Routing in the Oracle Space for Memory-constrained Large Language Model Inference. Proceedings of the 42nd International Conference on Machine Learning (ICML).

[2] Skliar, A., van Rozendaal, T., Lepert, R., Boinovski, T., Van Baalen, M., Nagel, M., Whatmough, P. N., & Bejnordi, B. E. (2025). Mixture of Cache-Conditional Experts for Efficient Mobile Device Inference. Transactions on Machine Learning Research (TMLR).

---

> ### Author Rebuttal · Authors · 2026-03-31
>
> Thank you for your professional review comments and suggestions. We have added results and will include them in revision.
>
> **Q1. Is it possible to provide a more direct comparison with Oracle-MoE or Mixture of Cache-Conditional Experts under the same architecture or comparable cache budget?**
>
> **A1.** Yes. These methods address the same bottleneck—poor temporal locality causing expert swapping and cache misses—but at different stages: Oracle-MoE changes routing architecture during pretraining, cache-conditional rerouting acts at inference time, while ReMoE is a post-training router-only adaptation method for existing open-source MoEs.
>
> Because Oracle-MoE requires architectural changes and full pretraining, a like-for-like comparison is difficult within the rebuttal window. To provide the most direct evidence, we instead reproduced a Skliar-style cache-aware inference heuristic on the same DeepSeek-V2-Lite model under the same cache setting (C=4, LRU). Results show that aggressively improving cache hits only through inference-time rerouting can severely damage model quality. ReMoE instead internalizes locality during fine-tuning and yields a better quality–locality trade-off.
>
> | Method               | β    | uHR ↑  | PPL ↓   |
> | -------------------- | ---- | ------ | ------- |
> | ReMoE (learned)      | 0.0  | 23.74% | 6.35    |
> | Baseline + heuristic | 1.0  | 41.66% | 10.60   |
> | Baseline + heuristic | 4.0  | 63.88% | 3607.92 |
>
> We also tested composability with a mild heuristic, which suggests that ReMoE is complementary to moderate cache-aware inference bias rather than a replacement.
>
> | Method                 | β    | uHR ↑  | PPL ↓ | Estimated TPS ↑ |
> | ---------------------- | ---- | ------ | ----- | --------------- |
> | Baseline + heuristic   | 0.5  | 32.07% | 6.97  | 2.0481          |
> | ReMoE + same heuristic | 0.5  | 34.07% | 6.51  | 2.1106          |
>
> **Q2. Could the authors include a “router CE-only continued fine-tuning” baseline to isolate the net contribution of ReMoE’s locality regularization?**
>
> A2. Yes. We added a CE-only router fine-tuning baseline: only optimized with standard LM cross-entropy, without ReMoE’s locality objective. CE-only does not reproduce the locality gain, and the same pattern appears in downstream evaluation. Thus, the gain cannot be explained by router adaptation alone; the locality-aware objective is necessary.
>
> | Metric         | CE-only | Baseline | ReMoE  |
> | -------------- | ------- | -------- | ------ |
> | EOR            | 0.2293  | 0.2730   | 0.3453 |
> | GSM8K (strict) | 36.92   | 38.89    | 38.13  |
> | GSM8K (flex)   | 37.23   | 39.04    | 38.36  |
> | HumanEval      | 28.05   | 26.83    | 29.27  |
> | MMLU           | 57.44   | 57.72    | 57.81  |
>
> Using vLLM (PR #37190; per-layer GPU LRU cache with CPU offloading), max-num-seqs=1, moe-expert-cache-size=6, and ShareGPT prompts (concurrency = 1), ReMoE improves inference efficiency:
>
> | Method    | Output token throughput (tok/s) ↑ | Mean TTFT (ms) ↓ | Mean TPOT (ms) ↓ | Avg. per-layer unique-expert hit rate ↑ |
> | --------- | --------------------------------- | ---------------- | ---------------- | --------------------------------------- |
> | Baseline  | 3.58                              | 769.23           | 254.31           | 39.4%                                   |
> | CE-only   | 2.95                              | 780.12           | 286.82           | 21.1%                                   |
> | **ReMoE** | **3.88**                          | **758.27**       | **242.99**       | **43.3%**                               |
>
> **Q3. There remains a surrogate gap between the theoretical analysis and the proposed methodology.**
>
> **A3.** We agree. Our theory is intended as supporting analysis, not a formal proof of the full method. The proposition relating EOR to expert fetches holds only under explicit assumptions such as C ≥ K, request-isolated decoding, and recency-based replacement. ReMoE also does not directly optimize discrete Top-K overlap; it optimizes distribution-level surrogates (reuse mass and trajectory regularization) that empirically encourage higher overlap. Likewise, Trust-KL is a soft anchor on routing distributions, not a hard guarantee of end-to-end invariance.
>
> We will revise the wording accordingly and present the theory as motivation and partial support for the surrogate design, with the main validation coming from experiments. At the same time, ablations show that the surrogate is meaningful in practice: removing Reuse reduces EOR from 0.3453 to 0.2831, while removing Trust increases EOR to 0.3877 but worsens language-model quality. This matches our intended interpretation: the locality terms drive reuse, while Trust-KL stabilizes the quality–locality trade-off rather than proving preservation formally.

---

> > ### Author Rebuttal · Reviewer_7YV6 · 2026-04-04
> >
> > Thank you for the detailed rebuttal. The authors have now addressed the main issues I raised in my original review. In particular, the added CE-only router fine-tuning baseline resolves my key concern about attribution: the locality and serving gains cannot be explained by generic continued router adaptation on the OpenHermes distribution alone. I also appreciate the clarification that the theoretical part should be interpreted as supporting analysis rather than a formal proof of end-to-end behavior preservation.
> >
> > The newly added serving-side results further strengthen the paper. From a systems perspective, I find this to be an especially attractive plug-and-play intervention: it preserves the inference graph, composes naturally with existing caching/prefetching/offloading stacks, and offers a near-free deployment-time performance win once the lightweight router-only adaptation is done.
> >
> > On the related-work side, I would still welcome a stronger final discussion of Oracle-MoE and Cache-Conditional Experts. That said, I must admit that, at present, there is still no directly comparable prior work for this exact plug-and-play post-training setting. Given this, I no longer view the comparison issue as a blocking weakness.
> >
> > Overall, the rebuttal has materially strengthened the paper and crossed the bar I had in mind when writing my original review. I am therefore updating my score from 3 to 5.

---

> > > ### Author Response · Authors · 2026-04-07
> > >
> > > Thank you for your time, your constructive feedback, and for reviewing our rebuttal. We are encouraged to hear that the newly added CE-only baseline and serving-side results successfully addressed your core concerns, and we deeply appreciate you raising the score.
> > >
> > > We agree with your final suggestion regarding the related literature. We will certainly include a more comprehensive and explicit discussion of Oracle-MoE and Mixture of Cache-Conditional Experts in the camera-ready version to better contextualize our approach.

---

### Official Review · Reviewer_bHo9 · 2026-03-10

**Soundness:** 2
**Presentation:** 3
**Significance:** 2
**Originality:** 2
**Overall Recommendation:** 3
**Confidence:** 5

**Summary:**

This work proposes ReMoE, a lightweight router-only fine-tuning approach that increases short-horizon expert reuse via temporal locality regularization while anchoring routing semantics with a frozen-reference trust loss, adding zero inference overhead. ReMoE improves
expert overlap and cache efficiency under standard policies, leading to higher throughput.

**Compliance With Llm Reviewing Policy:**

Affirmed.

**Final Justification:**

As mentioned in rebuttal ack.

**Key Questions For Authors:**

See weakness.

**Limitations:**

yes

**Strengths And Weaknesses:**

Strength:
1. This work is well-motivated, targeting the memory-constrained inference problem

2. The training is very lightweight, with only routers being fine-tuned

3. This work is overall well-presented with clear demonstrations.

Weakness:
1. This work lacks real-world on-device evaluations, with only offline cache simulation + proxy experiments. This makes the conclusions not very convincing.

2. Important baselines are missing. For example, 1) pretraining from scratch w. the routing locality loss; 2) expert pre-fetching for deeper layers

3. Important evaluation datasets are missing. For memory-constrained inference scenarios, tool-usage and instruction-following benchmarks are usually quite important, which are not presented in this paper.

---

> ### Author Rebuttal · Authors · 2026-03-31
>
> Thank you for your professional review comments and suggestions. We have added results and will include them in revision.
>
> **Q1. The work lacks real-world on-device evaluation.**
>
> **A1.** We added direct serving-side evaluation under expert offloading. We compare the original DeepSeek-V2-Lite checkpoint (baseline) with its router-only fine-tuned ReMoE counterpart under the same setup. On an RTX 3090 + Xeon Silver 4210 host, using vLLM (PR #37190; per-layer GPU LRU cache with CPU offloading), max-num-seqs=1, moe-expert-cache-size=6, and ShareGPT prompts (concurrency = 1):
>
> | Metric                          | ReMoE  | Baseline | Relative Change |
> | ------------------------------- | ------ | -------- | --------------- |
> | Output token throughput (tok/s) | 3.88   | 3.58     | +8.4%           |
> | Mean TTFT (ms)                  | 758.27 | 769.23   | +1.5%           |
> | Mean TPOT (ms)                  | 242.99 | 254.31   | +4.5%           |
>
> We also evaluate with llama.cpp (v8185) on a Jetson Orin NX 16GB, with the model stored on a NVMe SSD. Under identical settings (-np 1, -n 128, --mmap) on ShareGPT prompts:
>
> | Metric         | ReMoE   | Baseline | Relative Change |
> | -------------- | ------- | -------- | --------------- |
> | Mean TTFT (ms) | 3603.46 | 4246.59  | +15.1%          |
> | Mean TPOT (ms) | 232.73  | 501.12   | +53.6%          |
>
> ReMoE improves short-horizon routing locality so consecutive tokens reuse a smaller active expert set. In the decode phase of vLLM run, the average per-layer unique-expert hit rate rises from 39.4% to 43.3% (+3.9 pp), effectively reducing the overhead of expert switching.
>
> The absolute TPOT values are not directly comparable across platforms. The higher TPOT on RTX 3090/vLLM mainly reflects PCIe host-to-device expert transfers on cache misses, whereas Jetson Orin NX uses unified memory and avoids cross-bus copy overhead once experts are cached. Since Jetson’s SSD offloading path is much slower, cache misses are more expensive in the baseline, so ReMoE yields a much larger TPOT reduction there.
>
> **Q2: Some baselines are missing, such as (1) pretraining from scratch with the routing locality loss (2) expert pre-fetching for deeper layers.**
>
> **A2.** These are relevant, but they operate at different stages. ReMoE is a router-only post-training method, while prefetching is a runtime mechanism; they are complementary rather than interchangeable.
>
> For the training-side baseline, pretraining from scratch with locality-aware routing is not like-for-like for a lightweight post-training method because it requires a fundamentally different compute budget. We instead add a closer control, CE-Only, which continues fine-tuning the same model with standard next-token cross-entropy only, without our locality objective or Trust-KL anchoring.
>
> | Metric         | CE-only | Baseline | ReMoE  |
> | -------------- | ------- | -------- | ------ |
> | EOR            | 0.2293  | 0.2730   | 0.3453 |
> | GSM8K (strict) | 36.92   | 38.89    | 38.13  |
> | GSM8K (flex)   | 37.23   | 39.04    | 38.36  |
> | HumanEval      | 28.05   | 26.83    | 29.27  |
> | MMLU           | 57.44   | 57.72    | 57.81  |
>
> Under the vLLM offloading setup same as Q1, CE-only is also worse in serving and cache behavior: output throughput 2.95 tok/s, mean TTFT 780.12 ms, mean TPOT 286.82 ms, and average per-layer unique-expert hit rate 21.1%.
>
> For runtime prefetching, our point is that it is orthogonal to ReMoE. Prefetching acts at runtime, while ReMoE improves the routing trace itself. By reducing unique expert loads under LRU (e.g., #uMiss 1.0150M → 0.9746M at C=4, 0.8707M → 0.8068M at C=6), ReMoE makes the trace more predictable and thus more favorable to prefetching. This matters particularly in edge settings with limited I/O bandwidth, where expert transfer cannot be fully hidden behind compute. In addition, under small caches, incorrect prefetches are costly because they consume bandwidth and can evict actually needed experts.
>
> **Q3. Tool-usage and instruction-following benchmarks are not presented in this paper.**
>
> **A3.** We agree that instruction-following and tool-use evaluation are relevant. Although not included in the submitted manuscript, we additionally ran the instruction-following IFEval. ReMoE matches the baseline on prompt-level loose accuracy (17.93% vs. 17.93%), and the drop on prompt-level strict accuracy (17.19% → 16.08%) is within the reported standard error. This remains consistent with our claim that ReMoE preserves downstream capability while improving routing locality.
>
> For tool use, the issue is not relevance but testbed suitability. The public MoE backbones that are closest to our cache-limited local-deployment setting are officially positioned mainly as efficient/small MoE models rather than tool-calling benchmarks, whereas recent public MoE models with explicit tool-use support (e.g., Qwen3-30B-A3B) are materially larger and cannot be realistically run in our target edge scenario.

---

> > ### Author Rebuttal · Reviewer_bHo9 · 2026-04-02
> >
> > Thank the authors for their time and effort on the rebuttal. My questions have been partially addressed, but the following concerns remain:
> > 1. The motivation is still not very clear - in real-world we haven't seen (many) MoE models deployed on edge / resource-constrained devices. As also mentioned by Reviewer Egjq, it's hard to compete with dense counterparts.
> > 2. The throughput improvement seems marginal (eg. 3.88 vs. 3.58), which limits the practical usage.
> >
> > Therefore, I'd like to raise my score from 2 to 3.

---

> > > ### Author Response · Authors · 2026-04-07
> > >
> > > Thank you for your continued engagement and constructive feedback.
> > >
> > > **Q1. The motivation is still not very clear.**
> > >
> > > **A1.** We thank the reviewer for this point. Our motivation is that edge/resource-constrained MoE is no longer a purely hypothetical setting, and thus its deployment bottlenecks are timely to study.
> > >
> > > 1. [OPPO](https://www.oppo.com/en/newsroom/press/oppo-leads-ai-innovation-with-on-device-moe/) announced its claimed world’s first on-device MoE implementation, stating that it collaborated with leading chipset providers and that internal lab tests showed about 40% faster AI task processing together with improved energy efficiency. One of their chip providers [MediaTek](https://www.mediatek.com/products/smartphones/mediatek-dimensity-9400) states that its Dimensity 9400 / 9400+ platforms support hybrid MoE and latest MoE model support.
> > > 2. [NVIDIA](https://developer.nvidia.com/blog/build-next-gen-physical-ai-with-edge%E2%80%91first-llms-for-autonomous-vehicles-and-robotics/)’s latest TensorRT Edge-LLM release fully enables MoE support at the edge and targets resource-constrained platforms such as Jetson and DRIVE Thor, positioning MoE as a practical way to deliver higher-capability reasoning under strict latency and power budgets.
> > > 3. [Liquid AI](https://www.liquid.ai/blog/lfm2-8b-a1b-an-efficient-on-device-mixture-of-experts) released LFM2-8B-A1B, positioned for phones, tablets, and laptops; the official release and technical report show it matches the quality of 3–4B dense models while achieving 2.8–3.7× higher decode throughput than dense 4B baselines (e.g., Qwen3-4B) on real consumer CPU targets (Snapdragon SoC on Galaxy S24 Ultra and AMD Ryzen HX370).
> > > 4. [AllenAI](https://allenai.org/blog/olmoe-app) released OLMoE as a fully open-source on-device app; its official materials state that it runs completely on-device, requires no cloud connectivity, and reaches 41 tokens/s on an iPhone 16 Pro.
> > > 5. [Google](https://ai.google.dev/gemma/docs/core/model_card_4) states that Gemma 4’s 26B A4B MoE runs almost as fast as a 4B-parameter dense model while delivering quality within ~2% of the much larger 31B dense model across benchmarks, making it a strong fast-inference alternative to the dense 31B model.
> > >
> > > In summary, our motivation is to address a concrete and increasingly relevant deployment problem: when fine-grained MoE is deployed under limited fast memory, expert switching turns offloading into a first-order latency bottleneck. ReMoE targets this bottleneck directly by making routing more cache-friendly, while remaining complementary to system-level optimizations.
> > >
> > > **Q2. The throughput improvement seems marginal.**
> > >
> > > **A2.** Our target setting is memory-constrained, single-request edge decoding, where the more relevant user-facing metrics are TTFT and especially steady-state per-token latency (TPOT), rather than aggregate server throughput. In this case, “throughput” is essentially just another expression of TPOT (up to units), i.e., output tok/s is the inverse of per-token latency. Therefore, using server-style aggregate throughput as the primary lens here would create a metric mismatch: the practically relevant question is how quickly the next token reaches the user, not how many concurrent requests a server cluster could absorb.
> > >
> > > In addition to the Jetson results already reported in our previous rebuttal round, we further add the following workload-diverse edge evaluation. We evaluate on Jetson Orin NX 16GB + llama.cpp across three distinct workloads: ShareGPT (chat / open-ended generation), GSM8K (reasoning), and HumanEval (code generation):
> > >
> > > | Workload  | Mean TTFT (Base) | Mean TTFT (ReMoE) | TTFT Reduction | Mean TPOT (Base) | Mean TPOT (ReMoE) | TPOT Reduction | Decode Speed |
> > > | --------- | ---------------- | ----------------- | -------------- | ---------------- | ----------------- | -------------- | ------------ |
> > > | ShareGPT  | 7150.12 ms       | 5368.77 ms        | -24.9%         | 554.69 ms        | 306.27 ms         | -44.8%         | 1.81×        |
> > > | GSM8K     | 6041.65 ms       | 4736.70 ms        | -21.6%         | 613.73 ms        | 346.04 ms         | -43.6%         | 1.77×        |
> > > | HumanEval | 7185.78 ms       | 5233.11 ms        | -27.2%         | 672.68 ms        | 337.61 ms         | -49.8%         | 1.99×        |
> > >
> > > ReMoE improves short-horizon routing locality, so consecutive decoding steps reuse the same experts more often. In memory-constrained edge inference, expert misses are expensive because missing experts must be loaded from slower memory / storage tiers into limited fast memory. ReMoE reduces the number of such expert switches and reloads, thereby lowering the cumulative I/O cost during decoding. Since this I/O overhead is paid repeatedly token by token, the benefit appears most directly in steady-state TPOT, and also translates into TTFT gains.

---

### Decision · Program_Chairs · 2026-04-30

**Decision:**

Accept (regular)

**Comment:**

In this paper, the authors introduce ReMoE, a router fine-tuning framework designed to encourage expert reuse across adjacent tokens. The method is motivated by a practical systems issue: in memory-constrained MoE inference, frequent expert switching across tokens leads to substantial I/O overhead, as experts that are not resident in fast memory must be repeatedly loaded from slower storage. The core idea of the method is to introduce temporal-locality regularization so that nearby tokens are more likely to reuse the same experts, thereby reducing expert loading overhead and improving inference efficiency. To preserve the original model behavior, the authors additionally introduce a Trust-KL loss that keeps the fine-tuned routing distribution close to the pretrained one.

The reviews are mixed. All reviewers agree that the method is practical, especially for edge or memory-constrained deployment settings where inference often runs with batch size 1 and thus cannot benefit from parallelization across sequences. Another practical strength is that the training procedure is lightweight as only the router parameters are updated. Empirically, the method preserves downstream quality reasonably well, with only minor drops on GSM8K and slight improvements on HumanEval, while achieving substantial latency reductions in the edge-device setting (up to 40–50% TPOT reduction).

Nevertheless, some concerns were brought up during discussion period. Reviewer gQ3Q raises questions about reduced routing expressiveness and expert imbalance issue, while Reviewer Egjq and Reviewer bHo9 are less convinced by the broader motivation for deploying MoE models on edge devices. On the first point, the additional analysis during the discussion period clarifies that the reduction in routing diversity and the resulting expert imbalance are both quite limited, with only minor impact on model quality. On the second point, I find the rebuttal persuasive. The authors provide concrete examples from OPPO, NVIDIA, Liquid AI, AllenAI and Google to demonstrate that edge deployment of MoE models is no longer purely hypothetical but of practical in such settings.

Overall, this is a nice paper whose strengths outweigh its weaknesses, which offers a simple yet effective plug-in approach for deploying MoE models on edge devices. Therefore, I recommend acceptance and encourage the authors to incorporate the necessary discussion into the final version.